# Gut *Bacteroides* act in a microbial consortium to cause susceptibility to severe malaria

Rabindra K. Mandal[1], Anita Mandal[1], Joshua E. Denny[2], Ruth Namazii[3], Chandy C. John[1] & Nathan W. Schmidt [1,2] ✉

Malaria is caused by *Plasmodium* species and remains a significant cause of morbidity and mortality globally. Gut bacteria can influence the severity of malaria, but the contribution of specific bacteria to the risk of severe malaria is unknown. Here, multiomics approaches demonstrate that specific species of *Bacteroides* are causally linked to the risk of severe malaria. *Plasmodium yoelii* hyperparasitemia-resistant mice gavaged with murine-isolated *Bacteroides fragilis* develop *P. yoelii* hyperparasitemia. Moreover, *Bacteroides* are significantly more abundant in Ugandan children with severe malarial anemia than with asymptomatic *P. falciparum* infection. Human isolates of *Bacteroides caccae*, *Bacteroides uniformis*, and *Bacteroides ovatus* were able to cause susceptibility to severe malaria in mice. While monocolonization of germ-free mice with *Bacteroides* alone is insufficient to cause susceptibility to hyperparasitemia, meta-analysis across multiple studies support a main role for *Bacteroides* in susceptibility to severe malaria. Approaches that target gut *Bacteroides* present an opportunity to prevent severe malaria and associated deaths.

Malaria is an infectious disease caused by the bite of a female *Anopheles* mosquito carrying *Plasmodium* parasites that infect red blood cells of vertebrates. More than 150 species of *Plasmodium* are known, of which five (*P. falciparum*, *P. vivax*, *P. malariae*, *P. ovale*, *P. knowlesi*) can infect humans, and three (*P. yoelii*, *P. berghei*, *P. chabaudi*) can infect mice and are studied as rodent malaria models. Malaria is a significant global burden causing more than 627,000 deaths primarily associated with *Plasmodium falciparum* infection and 241 million cases in 2020 alone, with increasing trends, especially in children <5 years old in sub-Saharan Africa[1]. With the availability of only one approved malaria vaccine, exhibiting only ~30% efficacy, and a rise in resistance against antimalarial drugs and insecticides, there is an urgency to develop novel strategies that prevent life-threatening severe malaria[2–4].

Within the last decade, the gut microbiome has garnered attention as a novel risk factor for malaria outcomes in humans. The gut microbiome is an integral component of health and is associated with diseases like cancer, obesity, diabetes, AIDS, asthma, neurological disorders, and malaria, among others[5,6]. Yilmaz et al. showed that *Escherichia coli* O86:B7, a member of gut microbiota, produces a glycan, Galα1-3Galβ1-4GlcNAc-R (α-gal), that stimulates cross-reactive antibodies capable of conferring protection against malaria transmission from the site of a mosquito bite to the liver in both humans and mice[7]. Stool microbiota composition is associated with the risk of *P. falciparum* infection, where *Bifidobacterium* and *Streptococcus* were associated with decreased risk of infection but not with the risk of developing febrile malaria in humans[8]. Relative to uninfected children, children infected with *P. vivax* have increased *Bacteroides* but reduced *Prevotella* and *Clostridiaceae* in the gut, linking the relative abundance of these gut microbiota with malaria[9]. In another study, a significantly increased abundance of *Lactobacillus* at the genus level was associated

[1]Ryan White Center for Pediatric Infectious Diseases and Global Health, Herman B Wells Center for Pediatric Research, Department of Pediatrics, Indiana University School of Medicine, Indianapolis, IN, USA. [2]Department of Microbiology and Immunology, University of Louisville, Louisville, KY, USA. [3]Department of Paediatrics and Child Health, Makerere University, Kampala, Uganda. ✉e-mail: nwschmid@iu.edu

with mixed *P. falciparum* and *P. vivax* infection compared to uninfected children[10]. Of note, a limitation of these studies is the comparison of gut bacteria between *Plasmodium*-infected and uninfected children because it is not possible to know what the malaria outcomes would be in the uninfected children if they, too, were infected with *Plasmodium*. Importantly, we have addressed this limitation by comparing gut bacteria between Ugandan children with an asymptomatic *P. falciparum* infection to children with severe malarial anemia, which demonstrated significant differences in gut bacteria composition between children with an asymptomatic infection and severe malarial anemia[11].

Several studies have shown that baseline gut microbiota before *Plasmodium* infection is associated with the severity of malaria in mice[6,11–14]. Within these studies, transfer of gut microbiota to germ-free mice in a specific pathogen free facility demonstrated a causal role of gut microbiota as a risk factor for susceptibility to severity malaria. Gut microbiota composition is also linked with better pregnancy outcomes with *P. chabaudi* AS infection in mice[15]. Further, lung microbiota is correlated with the severity of malaria-associated acute respiratory distress syndrome (MA-ARDS), where *Plasmodium*-induced T cells stimulate the production of the anti-inflammatory cytokine IL-10 in the lung, leading to compromised control of lung microbiota[16]. Among these studies, gut microbiota that included *Bacteroides*, *Prevotella*, *Lachnospiraceae spps.*, *Lactobacillus*, among others, were shown to be associated with higher *Plasmodium* parasitemia[6,10–12,15,17,18]. Yet, none of the studies in mice or humans have delineated the causal role of a specific member or defined consortium of gut microbiota in severe malaria.

This study applied multiomics approaches to C57BL/6 mice, susceptible or resistant to *P. yoelii* 17XNL (Py) hyperparasitemia. Using shotgun metagenomics of gut (ceca), ceca and serum metabolomics, whole blood transcriptomics, and culturomics, we found that certain species of mouse and human gut *Bacteroides* were able to cause susceptibility to severe hyperparasitemia. Yet, monocolonization of germ-free mice in a gnotobiotic facility with *Bacteroides* species did not cause susceptibility to hyperparasitemia, indicating *Bacteroides* cause susceptibility to hyperparasitemia as part of an interaction/consortium effect.

## Results

### Hyperparasitemia-susceptible mice have enriched microbial metagenomic potential within the gut

We have previously shown that gut microbiota in mice from Charles River Laboratories and Envigo causes mice to be susceptible to hyperparasitemia following Py infection, while gut microbiota in mice from Taconic Biosciences and Jackson Laboratories causes mice to be resistant to hyperparasitemia following Py infection[6,11–14]. Transfer of ceca contents from these mice into germ-free mice established a causal role of gut microbiota towards susceptibility to Py hyperparasitemia and excluded genetic drift among these colonies of mice as a contributing factor. To identify specific members of gut microbiota responsible for susceptibility and resistance to Py hyperparasitemia, adult female specific-pathogen-free (SPF) C57BL/6 mice ($n = 75$) were obtained from four vendors, Charles River Laboratories (CR), Envigo (Env), Jackson Laboratory (Jax), and two isolated barrier units (IBU) from Taconic Biosciences (Tac) with differential susceptibility to Py hyperparasitemia[12], totaling five groups. After 1 week of rest, five mice from each group were infected with Py to track parasitemia with ceca content and ceca mucosal scraping obtained from the remaining ten mice from each group (Fig. 1a). Mice from Charles River Laboratories (CR), Envigo (Env), and Taconic Biosciences (Tac-S) were susceptible to hyperparasitemia (hereafter referred to as susceptible mice). In contrast, mice from the other Taconic Biosciences IBU (Tac-R) and Jackson Laboratory were resistant to hyperparasitemia (hereafter referred to as resistant mice, Fig. 1b). DNA was extracted from ceca

content and subjected to shotgun metagenomics sequencing. Quality microbial non-host reads were greater than 10 million per mouse (Fig. 1c).

Microbial reads were subjected to taxonomic assignment and analyses using four different bioinformatics pipelines, including K-mer (Clark), marker (MetaPhlan2), de novo assembly (MaxBin), and hybrid-based (CZID) approaches. The approach to use multiple pipelines was chosen to account for the strengths and weaknesses of diverse bioinformatic pipelines available for metagenome analysis. More than 99% of the microbial reads were from the bacterial kingdom (S Fig. 1A). These approaches identified overlapping and distinct bacterial species (S Fig. 1B–E). Alpha diversity measured using richness (observed species) and Shannon entropy were similar between Clark and MaxBin, while lowest for MetaPhlan2 and intermediate for CZID (Fig. 1d, e). Alpha diversity metrics between the groups of mice were variable between the different bioinformatic analyses (Fig. 1d, e). Gut beta diversity analysis performed using Bray-Curtis distance showed that all five groups had significantly different microbial compositions from each other using the four different bioinformatics pipelines (Fig. 1f–i). These data reinforce the known diversity of gut bacteria communities between and within vendors[12]. As such, functional analyses are also required to identify microbiota that impact malaria outcomes.

Over 2.9 million microbial genes were detected in the gut metagenome of all 50 commercial vendor-raised SPF mice, with 91.7% singletons at 95% sequence similarity. Env and CR mice had higher microbial genetic potential (gene content) than resistant mice (Fig. 1j). Whereas Tac-S mice had similar genetic potential as Jax mice, they had more genetic potential than Tac-R mice (Fig. 1j). Functional profiling of metagenomes at the pathway level using HUMAnN2 showed distinct clustering of CR and Jax mice with overlap between Env and Tac-S and a few shared samples between Tac-S and Tac-R (S Fig. 2A). Differential pathway analysis revealed the genomic potential for heme biosynthesis (aerobic), among others, were enriched in susceptible mice. In contrast, other specific pathways were enriched in resistant mice (S Fig. 2B). An alternative approach to analyze pathways was performed using GhostKOALA (S Fig. 2C)[19]. At a global level, similar functional categories were observed between CR and Tac-R mice, as indicated by the pie chart (S Fig. 2D, E). Consistent with the HUMAnN2 analysis, the number of genes belonging to KEGG modules alpha-Hemolysin/ cyclolysin transport system was >2.5 log2FC (fold change) in susceptible mice compared to resistant mice (S Fig. 2F). These data support the possibility that gut bacteria modulation of heme metabolism might be a mechanism by which gut bacteria impact susceptibility to Py hyperparasitemia.

Consistent with predicted metabolic potential using shogun metagenomics, global metabolomics profiling of ceca content from hyperparasitemia-susceptible mice (CR and Env) and hyperparasitemia-resistant mice (Tac-R and Jax) using ultrahigh performance liquid chromatography-tandem mass spectroscopy (UPLC-MS/MS) showed distinct clustering at metabolite and sub-pathway level (S Fig. 3A, B and E) indicating functional differences in the gut with differentially abundant metabolites and pathways (S Fig. 3C, D and F). However, serum metabolomics showed minimal differences between hyperparasitemia-resistant and -susceptible mice (S Fig. 4).

### Treatment with individual metagenome-resolved species were not able to modulate parasite burden

Linear discriminant analysis Effect Size (LEfSe)[20] and random forest classifier[21] were used to identify differentially abundant gut bacterial species that were taxonomically assigned using the four different pipelines between hyperparasitemia susceptible ($n = 30$) and resistant mice ($n = 20$). This multimodal approach identified multiple bacteria able to predict Py hyperparasitemia-resistant or -susceptible phenotype (Fig. 2a–f, and S Fig. 5). Although many bacteria were identified via one bioinformatic pipeline and statistical approach, several bacteria

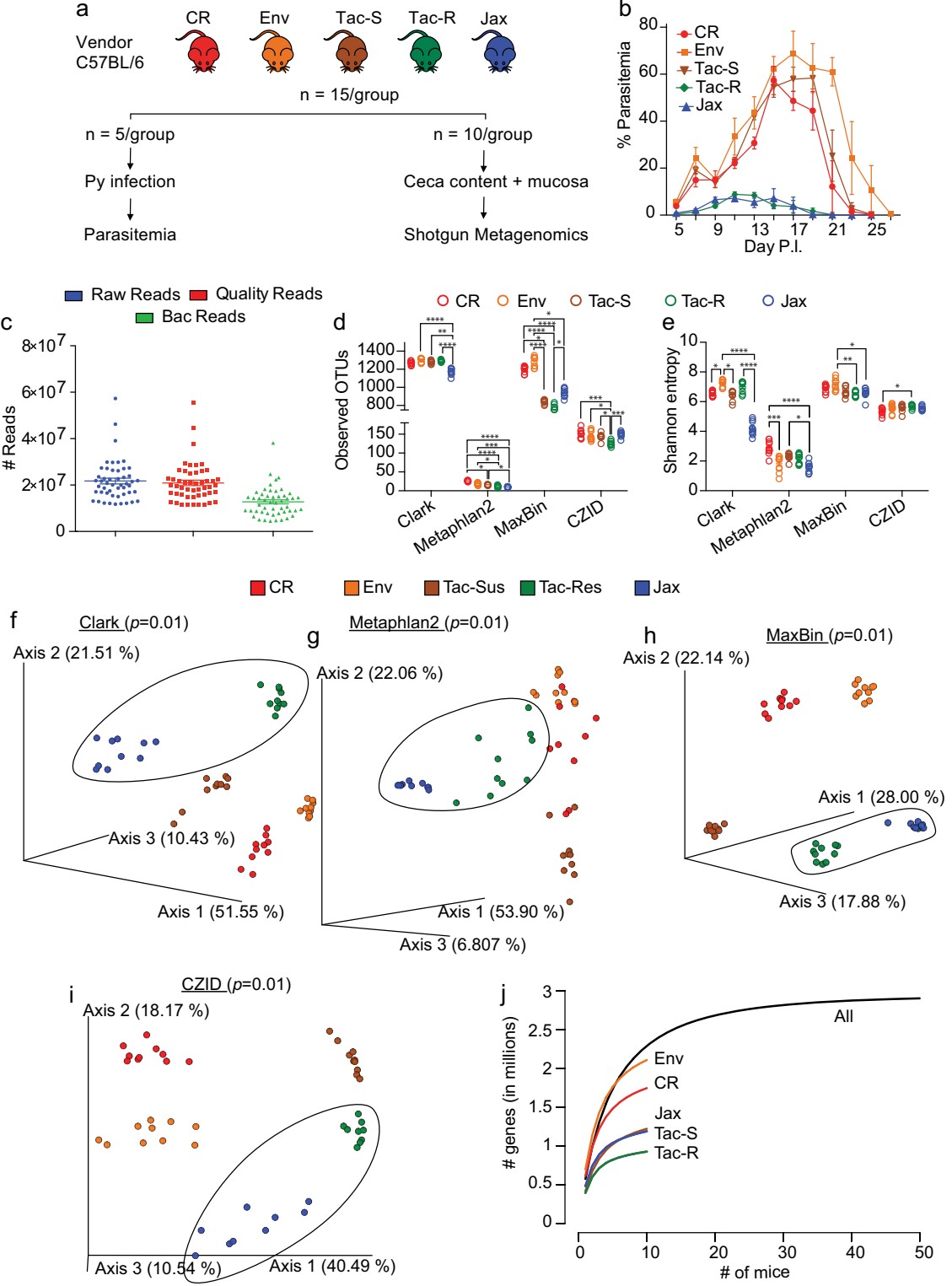

were identified via multiple bioinformatic pipelines and both statistical methods (Fig. 2a, b).

The ability of differentially abundant metagenome-resolved bacteria to modulate severe malaria in mice was prioritized and tested. *Micromonospora* species can produce significant secondary metabolites, including aminoglycoside antibiotics, gentamicin, and netamicin[22]. We previously demonstrated that gentamicin, including other antibiotics, could significantly decrease the severity of malaria in mice[11]. *Bifidobacterium* is a common probiotic that confers health benefits[23], *B. adolescentis* exerts anti-inflammatory properties in the

gut via inhibition of NF-κB activation and lipopolysaccharide secretion[24], and *Bifidobacterium* has been correlated with protection from *P. falciparum* infection and Py hyperparasitemia[6,18]. To determine the ability of these bacteria to mitigate severe malaria, hyperparasitemia-susceptible CR mice were gavaged daily from −7 to 7 days post-infection (p.i.) with *B. adolescentis* or *Micromonospora auranticia* (Fig. 2g). Parasitemia was not decreased by either of these bacteria (Fig. 2h). *Bacteroides fragilis* is immune-modulatory and identified as overly abundant in susceptible mice (Fig. 2b, e, and f; and S Fig. 5A and G). *Bacteroides fragilis* isolated from CR ceca on selective

**Fig. 1 | Shotgun metagenomics revealed distinct gut microbiota composition and genetic potential within and between the hyperparasitemia resistant and susceptible mice. a** C57BL/6 mice were acquired from four different vendors (N = 15/group): Charles River Laboratories (CR), Envigo (Env), Taconic Biosciences (Tac), and Jackson Laboratory (Jax). Mice from Taconic Biosciences were obtained from two different facilities with differential susceptibility to Py hyperparasitemia[12]. Five mice from each group were infected with Py while the remaining ten mice were sacrificed to collect ceca content along with mucosa scrapes for shotgun metagenomics. **b** Parasitemia curve of mice infected with Py. **c** Number of fastq reads after shotgun sequencing and quality control. **d** Alpha diversity measured using observed taxonomic units (OTUs) defined at species level. **e** Alpha diversity measure by Shannon entropy. **f** Beta diversity shown by Principal coordinate analysis (PCoA) plot using Clark output at species level with Bray-Curtis distance. **g** Beta diversity shown by PCoA plot using Metaphlan2 output at species level with Bray-Curtis distance. **h** Beta diversity shown by PCoA plot using MaxBin output at species level with Bray-Curtis distance. **i** Beta diversity shown by PCoA plot using CZID output at bin level with Bray-Curtis distance. **j** Number of unique genes detected at 95% sequence homology. All data are mean ± SE (standard error) unless explicitly stated. Bacterial diversity was performed on normalized data. Mice resistant to hyperparasitemia are encircled (**f**–**i**). Alpha diversity significance were calculated with Kruskal Wallis test and beta diversity significance by pairwise Permutational multivariate analysis of variance (PERMANOVA). *p < 0.05; **p < 0.01; ***p < 0.001; ****p < 0.0001. Exact p values are shown in Source Data file. Py: *Plasmodium yoelii* 17XNL; Tac-R and Tac-S: Taconic mice resistant and susceptible to hyperparasitemia to Py infection respectively. Source data are provided in the Source Data file.

media was continuously gavaged from −7 to 7 days post-Py infection to hyperparasitemia-resistant Tac-R mice but did not increase parasite burden in Tac-R mice (Fig. 2j, k). Although metagenomics analyses identified differentially abundant bacteria between hyperparasitemia-resistant and -susceptible mice, this singular approach did not yield bacteria that, following treatment, could confer resistance or susceptibility. These results highlight the complexity of gut microbiota interactions that culminate in resistance or susceptibility to Py hyperparasitemia and the need for additional investigation to identify bacteria that impact malaria outcomes.

## Whole blood transcriptomics and ceca metagenomics identified alpha toxin producing bacteria increase parasitemia

Alpha hemolysin are exotoxins secreted by bacteria like *Staphylococcus aureus* that can lyse erythrocytes[25]. One of the major complications of hemolysis is generation of harmful metabolites like labile heme[26] which are required by *Plasmodium* parasite as a metabolic cofactor[27]. We hypothesized that overabundance of the bacterial heme biosynthesis pathway and alpha-hemolysin/cyclolysin transport system in gut metagenome (S Fig. 2B and F) result in increased abundance of heme and overexpression of related pathways in the host that affects the host response to Py. To this end, whole blood bulk transcriptomics was performed in one group of Py hyperparasitemia-susceptible mice (CR) and two groups of Py hyperparasitemia-resistant mice (CR+Van[11] and Tac-R (Fig. 3a–c and S Fig. 6A–C). In support of the hypothesis, gene set enrichment analysis (GSEA) showed that, amongst other differences, the heme metabolism pathway was upregulated in Py hyperparasitemia-susceptible CR mice compared to Py hyperparasitemia-resistant Tac and CR+Van mice on day 0 before any hemolysis and Py infection (Fig. 3d). Yet, there were no observed differences in heme or hemoglobin between resistant and susceptible mice (S Fig. 7A and B). In support of prior studies showing gut microbiota modulate the humoral immune response to Py, hyperparasitemia-resistant mice exhibited upregulation of pathways involved in the formation of germinal center reactions (S Fig. 6D). To test the effect of heme metabolism on Py hyperparasitemia, the host (mouse) heme metabolism pathway was targeted with Tin protoporphyrin (SnPP; competitive inhibitor of heme oxygenase), hemopexin (binds heme), and hemin (iron-containing porphyrin). Although the serum level of heme was influenced by hemopexin and heme treatments (S Fig. 7C and D), none of these treatments influenced Py parasitemia in hyperparasitemia-resistant or -susceptible mice (Fig. 3e–g).

Subsequently, modulation of the heme metabolic pathway was performed by isolating alpha toxin-producing gut bacteria from ceca of naïve hyperparasitemia susceptible CR mice. We hypothesized alpha toxin-producing gut bacteria could result in the lysis of red blood cells and alter host heme metabolic pathways. To this end, CR ceca were cultured under anaerobic conditions for *Staphylococcus aureus* (*S. aureus*) and *Streptococcus* on selective agar (Aureus ChromoSelect Agar and Selective Streptococcus Agar, respectively) and nutrient-rich blood TSA agar that supports isolation of hemolytic bacteria. Bacteria were cultured on their respective broth medium from glycerol stock daily and gavaged consecutively from −5 to 7 days post Py infection. Interestingly, *S. aureus* was able to delay parasite clearance in Tac mice (Fig. 3h, i). These results identify a plausible role for alpha-toxin-producing bacteria in modulating Py parasite burden, but further studies are required.

## A guild of culturable gut bacteria increased parasite burden in hyperparasitemia-resistant mice

As the culturomics approach yielded a bacterium that plausibly could modulate Py parasite burden, this approach was expanded to include different types of nutrient-rich agar. The hypothesis was that particular media would uniquely select/enrich a guild of gut bacteria that exacerbate severe malaria in mice. It is worth noting that culturomics will not allow the growth of all gut bacteria. As such, any unculturable bacteria that contribute to modulating the severity of malaria will not be identified. The decision to focus on gut bacteria linked to susceptibility to Py hyperparasitemia rather than resistance was based on several observations. First, CR but not Tac-R mice treated with antibiotics exhibit reduced parasitemia[11]. Second, the transfer of ceca content from CR mice into Tac-R mice conferred susceptibility to Py hyperparasitemia, but the transfer of Tac-R ceca content into CR mice did not confer resistance to Py hyperparasitemia[11]. Third, germ-free mice infected with Py exhibited low parasitemia similar to Tac-R mice (data not shown).

Ceca contents from CR mice were cultured on six different types of nutrient-rich agar to identify bacteria responsible for susceptibility to Py hyperparasitemia. Naïve hyperparasitemia susceptible CR ceca were pooled from 5 mice and serially plated on LB, MRS, RCM, TSA, Wilkins-Chalgren, and BHI and incubated in an anaerobic chamber at 37 °C for 48 hours. Bacteria were harvested from all the serially diluted plates, and glycerol stocks were made (Fig. 4a). Glycerol stocks were cultured, centrifuged, and the bacterial pellet was resuspended in 1 ml sterile saline daily. Resuspended bacteria were gavaged consecutively from −5 to 7 days post-Py infection to Tac-R mice (Fig. 4b, c). Only BHI culturable bacteria (S Fig. 8A) significantly increased parasite burden in Tac-R mice (Fig. 4d–f). Although a significant increase in parasitemia was observed, it remained 3- to 5-fold lower than hyperparasitemia susceptible CR mice.

Two additional murine models of resistance to Py hyperparasitemia were tested to characterize the impact of CR ceca BHI (CrBHI) grown bacteria in severe malaria. First, hyperparasitemia-resistant Jax mice were gavaged with CrBHI bacteria. Peak parasitemia reached ~20% and significantly increased parasite load compared to untreated Jax mice (Fig. 4g, h). Second, hyperparasitemia-resistant CR+Van mice were gavaged with CrBHI bacteria (Fig. 4i). In this model, CrBHI gavage was able to significantly increase parasite burden in CR+Van mice (CrBHI→CR+Van) compared to saline-treated CR+Van mice to a parasite burden comparable to untreated CR mice (Fig. 4j, k). Gut microbiota analysis using MVRSION sequencing showed more similar gut

**a**

| Resistant Bacteria | Method |
|---|---|
| *Micromonospora aurantiaca* | Clark, rf |
| *Alteromonas macleodii* | Clark, rf |
| *Bifidobacterium adolescentis* | Clark, rf |
| *Ruminococcus bromii* | MaxBin, rf, lefse |
| *Clostridium sp. SY8519* | MaxBin, rf |
| Butyrate producing bacterium SS3/4 | MaxBin, rf |
| *Lactobacillus johnsonii* | Clark, rf; Metaphlan2, lefse |
| *Akkermansia muciniphila* | Clark, MaxBin, and CZID, rf; Metaphlan2, lefse |
| *Bacteroides thetaiotamicron* | Metaphlan2, lefse; CZID, rf |

**b**

| Susceptible Bacteria | Method |
|---|---|
| *LACHNOSPIRACEAE BACTERIUM 3_1_57FAA_CT1* | MaxBin, rf |
| *CANDIDATUS ARTHROMITUS SP. SFB CO* | MaxBin, rf |
| *Lacotacillus reuteri* | Metaphaln2, lefse; Clark and CZID, rf |
| *Lactobacillus crispatus* | Clark, rf |
| *Bacteroides fragilis* | Clark and CZID, rf |
| *Bacteroides acidifaciens* | Clark, lefse |

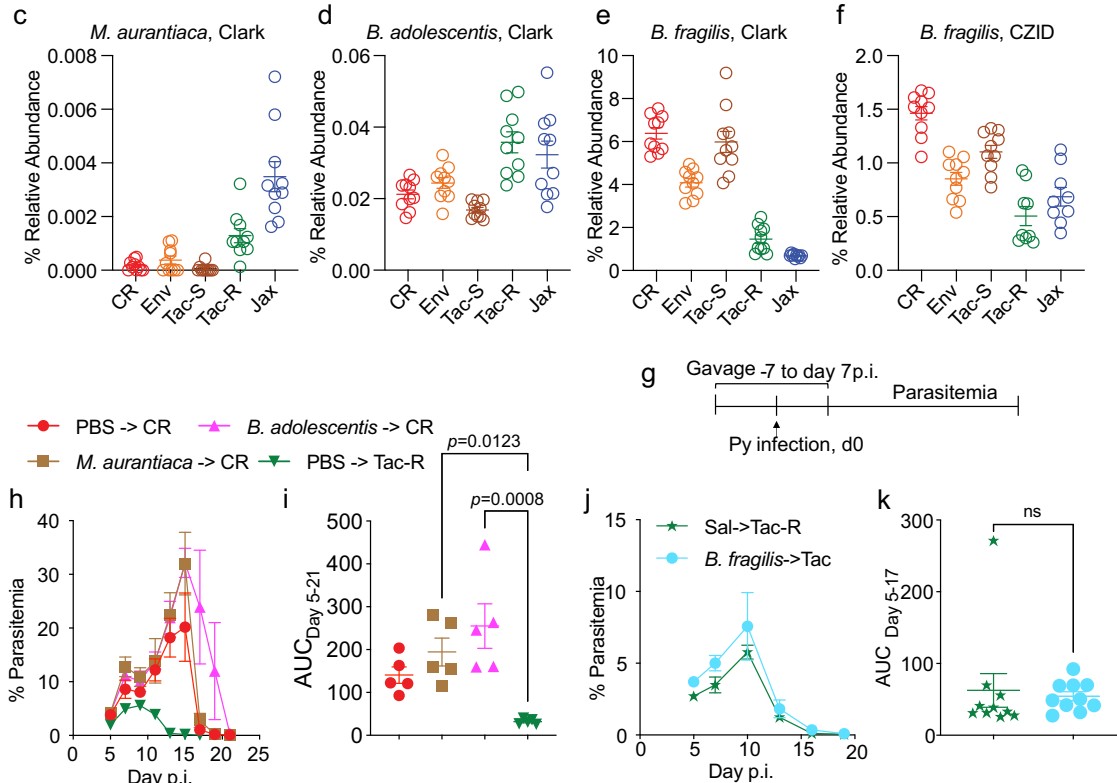

**Fig. 2 | Differentially abundant gut bacteria at species level in mice with differential Py parasite burden. a** Bacteria significantly abundant in mice resistant to hyperparasitemia. **b** Bacteria significantly abundant in mice susceptible to hyperparasitemia. **c** Relative abundance of *M. aurantiaca* identified by Clark pipeline. **d** Relative abundance of *Bifidobacterium adolescentis* identified by Clark pipeline. **e** Relative abundance of *Bacteroides fragilis* identified by Clark pipeline. **f** Relative abundance of *B. fragilis* identified by CZID pipeline. **g** Mice were gavaged daily with bacterial culture from day −7 to 7 Py infection. **h** Parasitemia curve of mice gavaged with *B. adolescentis* and *M. aurantiaca*. **i** Area under the parasitemia curve (AUC) analysis. **j** Parasitemia curve of mice gavaged with *B. fragilis* isolated from CR ceca. **k** AUC analysis. All data are mean ± SE (standard error). Two independent experiments were performed with N = 5 per group (**j**) except (**h**). AUC analyses were performed using one-way ANOVA. ns: non-significant, LEfSe: Linear discriminant analysis effect size, rf: random forest. Source data are provided in the Source Data file.

microbiota composition between CR mice and CrBHI→CR+Van indicated by alpha and beta diversity analysis and heatmap on day 9 post-infection (Fig. 4l–o, S Fig. 8B). Moreover, results showed 87.5% overlap between CR and CrBHI→CR mice post-infection on day 9 (S Fig. 8C). These results demonstrate that a guild of CrBHI bacteria contributes to susceptibility to Py hyperparasitemia.

## Mouse gut *Bacteroides fragilis* are capable of causing susceptibility to severe malaria

With the ability to culture a consortium of bacteria that caused susceptibility to severe malaria, we set out to identify specific bacterial species that caused susceptibility to Py hyperparasitemia. Shotgun metagenomics of culturable bacteria identified more than

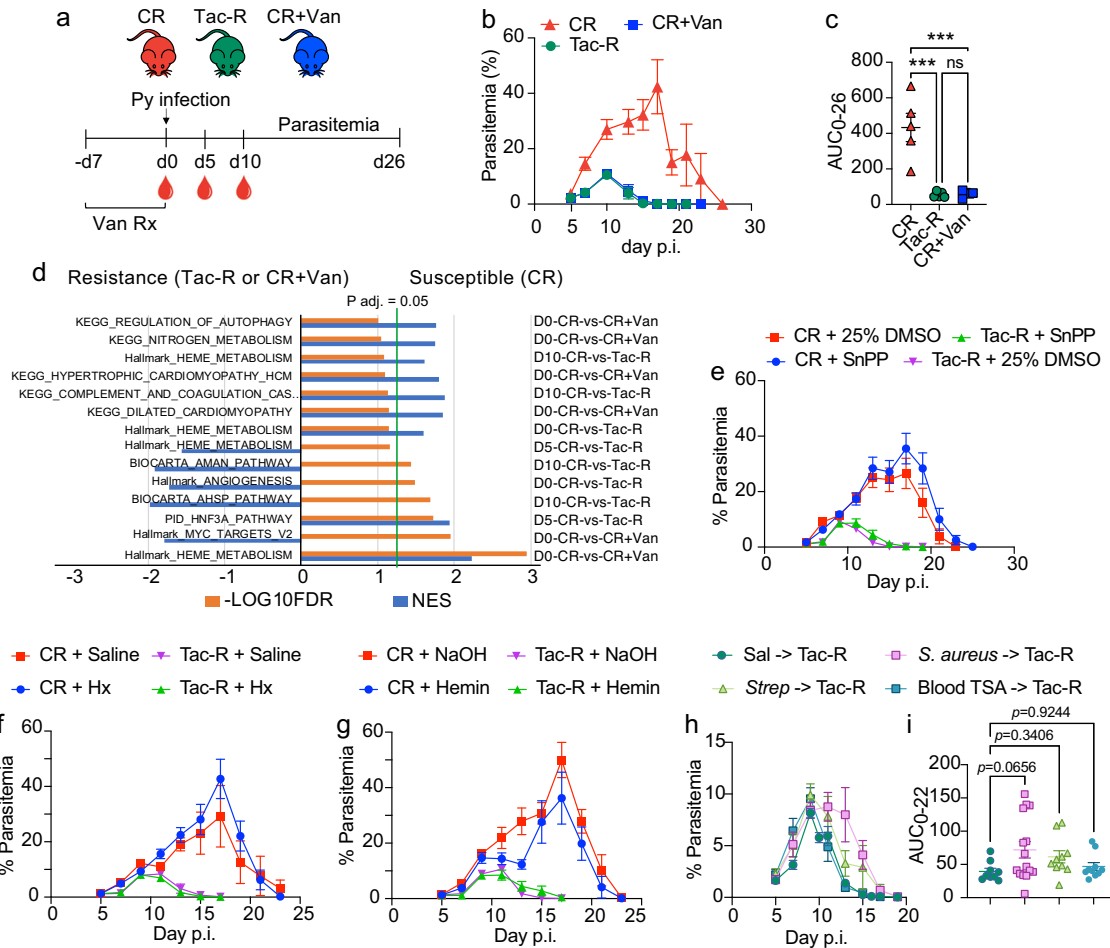

**Fig. 3 | Whole blood transcriptomics identify alpha toxin producing bacteria increase Py parasite burden. a** C57BL/6 CR mice were treated with vancomycin (CR+Van) in drinking water for 1 week prior to Py infection. Blood was collected retro-orbitally in RNAlater on day 0, 5, and 10 p.i. from CR, CR+Van and Tac-R and subjected to metatranscriptomics. **b** Parasitemia curve of Py infected mice. **c** AUC analysis. $N = 5$/group. CR vs. Tac-R, $p = 0.0002$; CR vs. CR+Van, $p = 0.0003$. **d** Gene set enrichment analysis (GSEA) using Hallmark gene sets, KEGG, BIOCARTA, and PID pathways. Top significant results are merged. Horizontal bar in orange is p adj. value and blue are normalized enrichment score (NES). $N = 3$ mice for day 0 and

$n = 4$ mice for day 5 and 10 post infection. **e** Parasitemia curve of CR and Tac-R mice treated with Tin protoporphyrin (SnPP). **f** Parasitemia curve of CR and Tac-R mice treated with hemopexin (Hx). **g** Parasitemia curve of CR and Tac-R mice treated with hemin. **h** Parasitemia curve of Tac-R mice gavaged with *Streptococcus* (Strep), *Staphylococcus aureus* (*S. aureus*) and gut bacteria harvested on Blood TSA agar. **i** AUC analysis. All data are mean ± SE (standard error). AUC were analyzed using one-way ANOVA. Data presented are at least from two independent experiment with $n = 5$/group in each experiment (**e**–**i**). Source data are provided in the Source Data file.

20 different species of bacteria competent to grow in CrBHI glycerol stock and inoculum (gavage input) prepared after additional 48 hours of incubation under anaerobic conditions (S Fig. 8A). To prioritize specific bacteria and increase the likelihood of success in identifying causal culturable gut bacterial species in CrBHI, CR+Van mice were gavaged with CrBHI (CrBHI→CR+Van) for eight consecutive days. Ceca were pooled from CrBHI→CR+Van mice and control saline-treated CR+Van mice, serially diluted, and anaerobically cultured on BHI agar. Glycerol stocks were made from harvested bacteria. Stocks were grown in BHI medium for 48 hours at 37 °C and centrifuged to collect bacteria. DNA was extracted and subjected to deep shotgun sequencing (Fig. 5a). On average, sequencing yielded ~140 million PE number reads per sample (Fig. 5b). BHI culture of cecal bacteria from CR+Van had significantly different alpha and beta diversity compared to CrBHI→CR+Van (Fig. 5c–f). Hyperparasitemia susceptible CrBHI→CR+Van mice ceca cultured on BHI had a significant overabundance of several *Bacteroides* species, *B. uniformis*, *B. faecis*, *B. intestinalis*, *B. vulgatus*, *B. thetaiotaomicron*, *B. stercoris* among others (Fig. 5f, g).

To obtain pure isolates of these *Bacteroides* species, single pure bacterial colonies were isolated from CrBHI glycerol stock on

BHI agar plates and genotyped using MALDI-TOF and 16S rRNA full-length Sanger sequencing. Unfortunately, among the 52 colonies selected on BHI, most were *E. coli* or *Enterococcus faecalis* (S Table1), which can be partially explained by the overabundance of *E. coli* and *E. faecalis* in the CrBHI glycerol stock (S Fig. 8A). Next, CrBHI was selected on MRS agar. Again, these efforts were unsuccessful as they only yielded isolates of *Lactobacillus intestinalis*, *L. murinus*, and *L. reuteri* (S Table 1), which were also detected by shotgun metagenome analysis of CrBHI glycerol stock (S Fig. 8A).

Previously, we showed that *B. fragilis* was significantly abundant in the ceca of Py hyperparasitemia susceptible mice (CR, Env, and Tac-S) (Fig. 2b, e, and f). Of note, among the species of *Bacteroides* genera, *B. fragilis* is the species for which there is a commercially available selective media. Nonetheless, multiple gavages with *B. fragilis* isolated using *B. fragilis* selective media from CR ceca to Tac-R mice did not increase Py parasitemia (Fig. 2j, k). Yet, the observation that CrBHI treatment of Tac-R mice yielded only a modest increase in Py parasitemia while CrBHI treatment of CR+Van mice yielded a substantial increase in parasitemia led to the hypothesis that *B. fragilis* treatment of CR+Van mice would have a

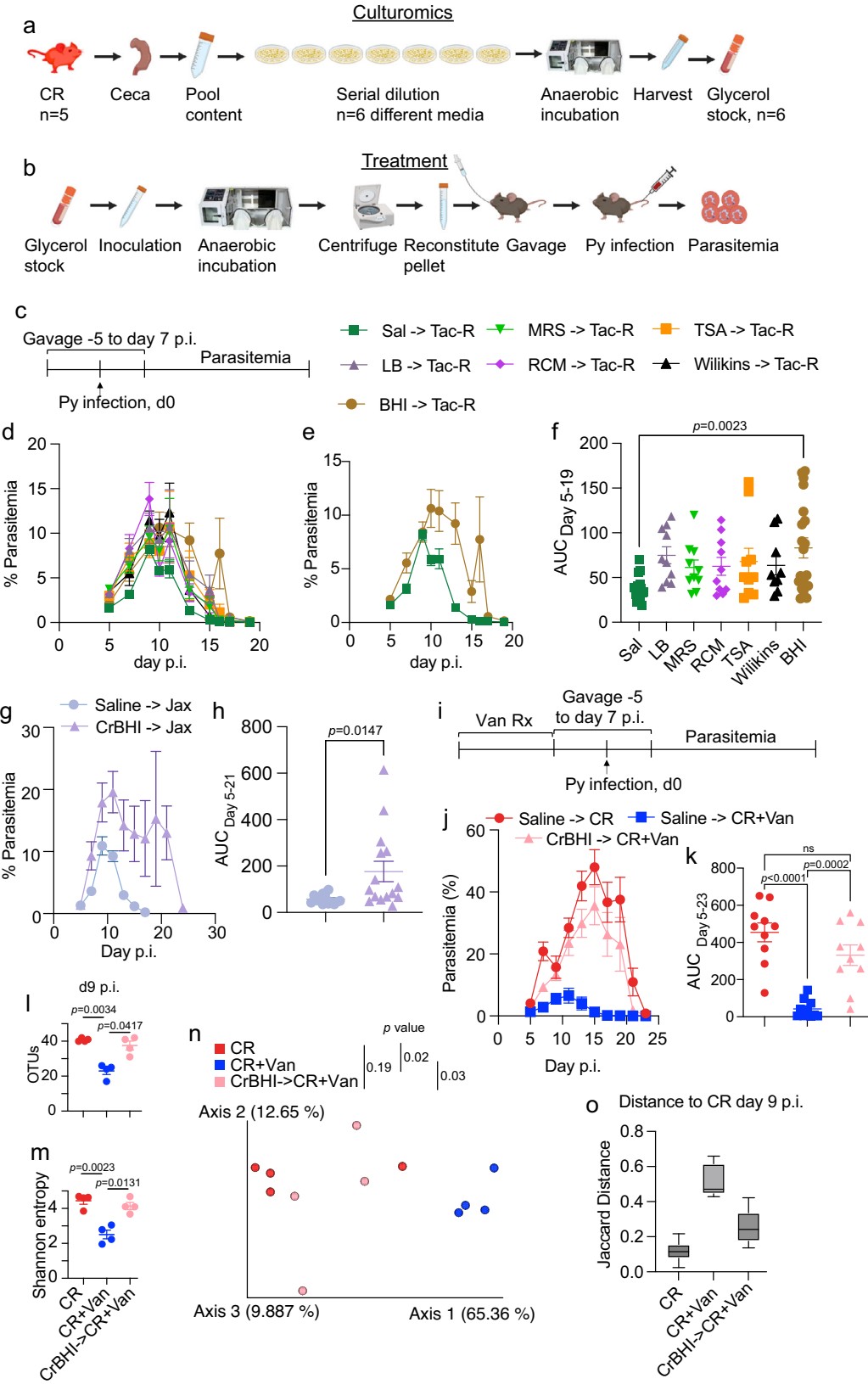

more pronounced effect. To test this possibility, CR+Van mice were gavaged with *B. fragilis* after cessation of vancomycin water for eight consecutive days before Py infection (Fig. 5h). CR+Van mice gavaged with murine-isolated *B. fragilis* had significantly higher parasite burden than control CR+Van mice with no difference in parasitemia between B. *fragilis*→CR+Van and control saline→CR.

(Fig. 5i, j). Despite the overabundance of *E. coli* in the CrBHI harvest, treating CR+Van mice with *E. coli* isolated from the CrBHI glycerol stock did not influence parasitemia (Fig. 5h, k, l), demonstrating that not all CrBHI bacteria cause susceptibility to Py hyperparasitemia. These data support that *B. fragilis*, can cause susceptibility to Py hyperparasitemia.

**Fig. 4 | A consortium of CR ceca culturable bacteria cause severe malaria. a** Ceca from five naïve CR mice were pooled, serially diluted on six different types of nutrient rich media. Bacteria grown in anaerobic condition were harvested and stored in 50% glycerol. **b** Glycerol stocks were inoculated in respective medium and incubated anaerobically. Bacterial pellet was resuspended in 1 ml saline after centrifugation. 200 μl was gavaged daily per mouse. **c** Mice were gavaged from −5 to day 7 p.i. **d** Parasitemia curve of Tac-R mice gavaged with bacteria harvested from CR ceca. **e** Parasitemia curve of control Tac-R and BHI gavaged to Tac-R (BHI→Tac-R). **f** AUC analysis. **g** Parasitemia curve of C57BL/6 mice from Jackson (Jax) gavaged with bacteria harvested from CR ceca on BHI media (CrBHI). **h** AUC analysis. **i** CR mice treated with 2 weeks of vancomycin followed by BHI gavage (CrBHI→CR+Van). **j** Parasitemia curve of CR, CR+Van, and CrBHI→CR+Van. **k** AUC analysis. **l** Alpha diversity measured using OTUs defined at species level. **m** Alpha diversity measures using Shannon entropy defined at species level. **n** Beta diversity shown by PCoA plot using Jaccard distance at species level. **o** Jaccard distance of CR mice to CR +Van and CrBHI→CR+Van at d9 post infection. Box plot whiskers represent minimum and maximum value and horizontal line inside the box is mean. All data are mean ± SE (standard error). AUC were analyzed using one-way ANOVA. Data presented are at least from two independent experiment with $n = 5$/group (**i–n**). ns: non-significant; Sal: saline; LB: Luria Bertani medium; MRS: De Man, Rogosa and Sharpe medium; RCM: Reinforced Clostridial medium; TSA: Trypticase Soy Agar; Wilkins: Wilkins-Chalgren Anaerobe Agar; and BHI: Brain Heart Infusion Agar. Source data are provided in the Source Data file. **a, b** Created with BioRender.com.

## Multiple *Bacteroides* enriched in the stool of children with severe malaria anemia exacerbates severe malaria in the murine malaria model

To extend the significance of our findings to severe malaria in humans, we evaluated the relative abundance of *Bacteroides* in Ugandan children with differential malaria outcomes. We previously reported that Ugandan children with severe malarial anemia (SMA) had significantly different stool bacteria communities than children with asymptomatic *P. falciparum* infection[11]. Machine learning using random forest showed that, among *Bacteroides* spp., *B. caccae* was a top predictor of malaria severity in Ugandan children to classify severe malaria anemia (SMA), healthy community children that were *P. falciparum* positive (asymptomatic; Pf-pos), and healthy community children that were *P. falciparum* negative[11]. Additional analysis of those samples revealed a higher abundance of *Bacteroides* at the genus level (Fig. 6a, b) and B. *caccae, B. fragilis, B. uniformis, B. vulgatus*, and *B. thetaiotaomicron* at the species level (Fig. 6c–i) in the stool of children with SMA compared to children with asymptomatic *P. falciparum* (Pf-pos) infection.

As with many human gut microbiome analyses, an important limitation in the identification of increased *Bacteroides* in children with SMA compared to children with asymptomatic *P. falciparum* infection is that these are correlative observations. Additionally, *Plasmodium* infections have been shown in murine malaria to change gut bacteria[17, 28]. However, Kenyan infants infected with *P. falciparum* showed no changes in gut bacteria while comparing stool samples collected within 2 weeks before and 2 weeks after infection[29]. Furthermore, it was shown that transfer of ceca content following Py-induced changes in gut bacteria from Py hyperparasitemia-susceptible and -resistant mice into germ-free mice did not alter the ability of gut microbiota from those mice to confer susceptibility or resistance, respectively, to hyperparasitemia following Py infection[17]. To move beyond the correlative observation and test the potential contribution of these *Bacteroides* species towards differential malaria outcomes, human isolates of *B. ovatus, B. caccae, B. uniformis*, and *B. thetaiotaomicron* were acquired from ATCC. CR+Van mice were gavaged for eight consecutive days after 2 weeks of vancomycin treatment in drinking water (Fig. 6j). Intriguingly, all *Bacteroides* species, except *B. thetaiotaomicron*, were able to significantly increase Py parasite burden compared to CR+Van (Fig. 6k–n). Although *B. ovatus* was not significantly different between SMA and Pf-pos, it was able to increase susceptibility to Py hyperparasitemia significantly. In contrast, although *B. thetaiotaomicron* was overly abundant in children with SMA compared to Pf-pos, treatment did not increase the Py parasite burden in mice. Of note, *B. thetaiotaomicron* was the only species of *Bacteroides* genus overly abundant (more than 20% relative abundance) in Jax mice, which are resistant to Py hyperparasitemia (S Fig. 9). The role of *B. vulgatus* and *B. dorei* in severe malaria is yet to be determined. Collectively, the data demonstrate a causal role of *Bacteroides* towards susceptibility to Py hyperparasitemia with implications of *Bacteroides* contributing to differential malaria outcomes in humans.

Consistent with the differential effect of *B. thetaiotaomicron* on parasite burden compared to the other *Bacteroides* species, gut bacteria analysis using MVRSION showed distinct groupings of hyperparasitemia-resistant and -susceptible mice (S Fig. 10A–E). The gut microbiota composition was more similar among hyperparasitemia susceptible mice, i.e., control CR mice and CR+Van mice gavaged singly with *B. caccae, B. uniformis, B. ovatus*, and *B. fragilis* (S Fig. 10A–E). On the contrary, Py hyperparasitemia-resistant groups (i.e., CR+Van and CR+Van mice gavaged with *B. thetaiotaomicron*) had the most dissimilar gut microbiota compared to control CR mice (S Fig. 10C and D). Of note, *B. thetaiotaomicron* is more distantly related to the other *Bacteroides* species based on 16S rRNA sequence homology (S Fig. 10F).

## *Bacteroides* species interact with other members of the gut microbiota to cause susceptibility to severe malaria

Finally, the individual or consortium (interaction) effect of gut *Bacteroides* in causing susceptibility to Py hyperparasitemia were tested. Ceca microbiota transplant from Tac and CR mice to germ-free mice showed that malaria severity was phenocopied in the reconstituted germ-free mice (Fig. 7a, b). These data demonstrate reconstitution of germ-free mice provides a model to investigate gut microbiota effects towards malaria outcomes, specifically the sufficiency of individual bacteria or requirement of a consortium of bacteria.

Human isolate *B. caccae* was selected for these studies because it was a top predictor of SMA in Ugandan children[11] and was shown to increase Py parasite burden in CR+Van mice (Fig. 6k, l). Human isolate *B. thetaiotaomicron* and mouse isolate *E. coli* were selected as control bacteria that do not cause an increase in Py parasite burden in CR+Van mice (Fig. 5k, l; and Fig. 6m, n). Germ-free mice were left untreated or individually colonized with *B. caccae, B. thetaiotaomicron*, and *E. coli* before Py infection (Fig. 7e). Germ-free mice were resistant to Py hyperparasitemia (peak parasitemia of ~10%; Fig. 7f, g), supporting that specific bacteria cause susceptibility to Py hyperparasitemia. As expected, germ-free mice gavaged with *B. thetaiotaomicron* and *E. coli* were resistant to Py hyperparasitemia, with parasite burdens even lower than the germ-free mice (Fig. 7f, g). Germ-free mice colonized with human isolate *B. caccae* were also resistant to Py hyperparasitemia and had lower parasitemia than germ-free mice (Fig. 7f, g), which contrasts hyperparasitemia in CR+Van mice treated with *B. caccae* (Fig. 5i, j). These data demonstrate *B. caccae* are not sufficient to cause susceptibility to Py hyperparasitemia; instead, *B. caccae* exerts this effect as part of a consortium of microbiota. Further supporting this hypothesis, *B. fragilis* treatment of Tac-R mice showed low parasitemia (Fig. 2j, k), whereas *B. fragilis* treatment of CR+Van mice exhibited susceptibility to Py hyperparasitemia (Fig. 5).

Based on meta-analysis including our previous publications, Mandal et al. (2020) and Mandal et al. (2021) and current work, we manually investigated the presence and absence of bacteria interacting with each other to identify the consortium of bacteria associated with susceptibility to Py hyperparasitemia[11,12]. The presence or absence of *Bacteroides* species was insufficient to explain differential susceptibility to Py hyperparasitemia. For example, "CR+Genta" and "CR +Amp" mice are resistant to Py hyperparasitemia despite the presence

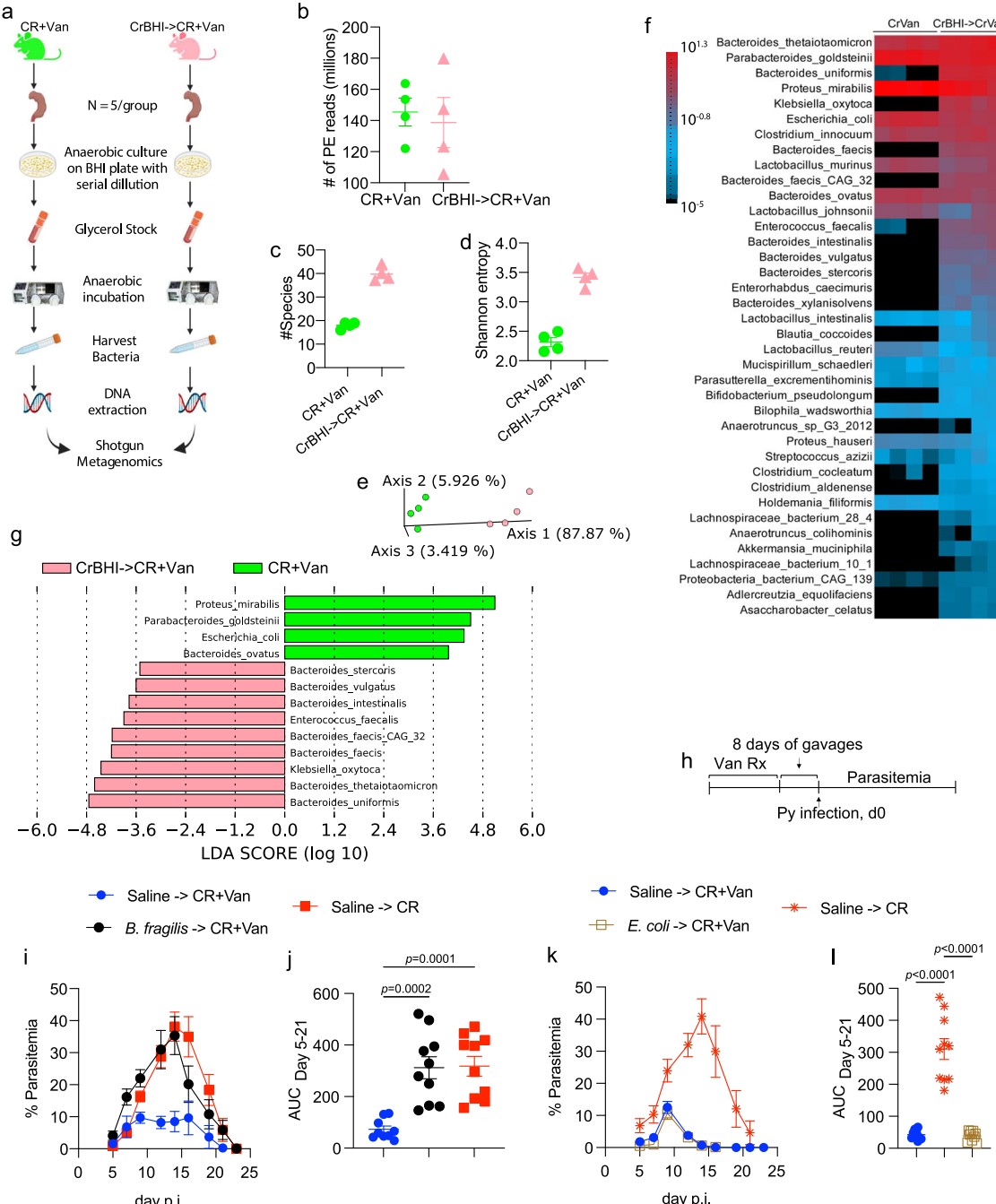

**Fig. 5 | Murine isolate *Bacteroides fragilis* have causal association with severe malaria. a** C57BL/6 mice from CR treated with vancomycin for 2 weeks (CR+Van) and gavaged with CrBHI for eight consecutive days after cessation of vancomycin (CrBHI→CR+Van). Naïve mice (*n* = 5/group) were sacrificed and ceca content were processed for culturing as described in previous section. **b** Number of paired end (PE) reads after quality control of cultured bacteria from (**a**). **c** Number of species detected. **d** Alpha diversity measured using Shannon entropy at species level. **e** Beta diversity shown by PCoA plot using Bray-Curtis distance at species level. **f** Heatmap showing abundance of bacterial taxonomy. **g** Differentially abundant bacteria identified using LEfSe. *N* = 4 independent bacterial culture (**b**–**g**). **h** C57BL/6 mice from CR mice were treated with vancomycin for 2 weeks followed by daily gavages with single bacterial species. **i** Parasitemia curve of mice gavaged with mouse isolate *B. fragilis*. **j** AUC analysis. **k** Parasitemia curve of mice gavaged with mouse isolate *E. coli*. **l** AUC analysis. All data are mean ± SE (standard error). AUC were analyzed using one-way ANOVA. Data presented are from two independent experiments with *n* = 5/group in each experiment (**i**–**l**). Source data are provided in the Source Data file. **a** Created with BioRender.com.

of *Bacteroides*; however, *Bacteroides* species were present in all Py hyperparasitemia groups (Fig. 7f). A meta-analysis across multiple studies identified a consortium of *Bacteroides* species (excluding *B. thetaiotaomicron*), *Lactobacillus reuteri*, *Bilophila*, *Parasutterella*, and *Alistipes* are frequently present in Py hyperparasitemia-susceptible groups (Fig. 7f). Of note, *Bilophila*, *Parasutterella*, and *Alistipes* species were not detected in the CrBHI harvest (S Fig. 6A), which might explain

the inability of CrBHI to exacerbate severe malaria in Tac-R and Jax mice because the entire combination of these bacteria (*Bilophila*, *Parasutterella*, *Alistipes* genera, and *Lactobacillus reuteri*) are missing from Tac-R and Jax, respectively. Additional interactions likely exist among gut bacteria, as indicated by the significant bacterial interaction network within CR gut to cause severe malaria, where *Bacteroides* species positively interact with *Prevotella*, *Porphyromonas*,

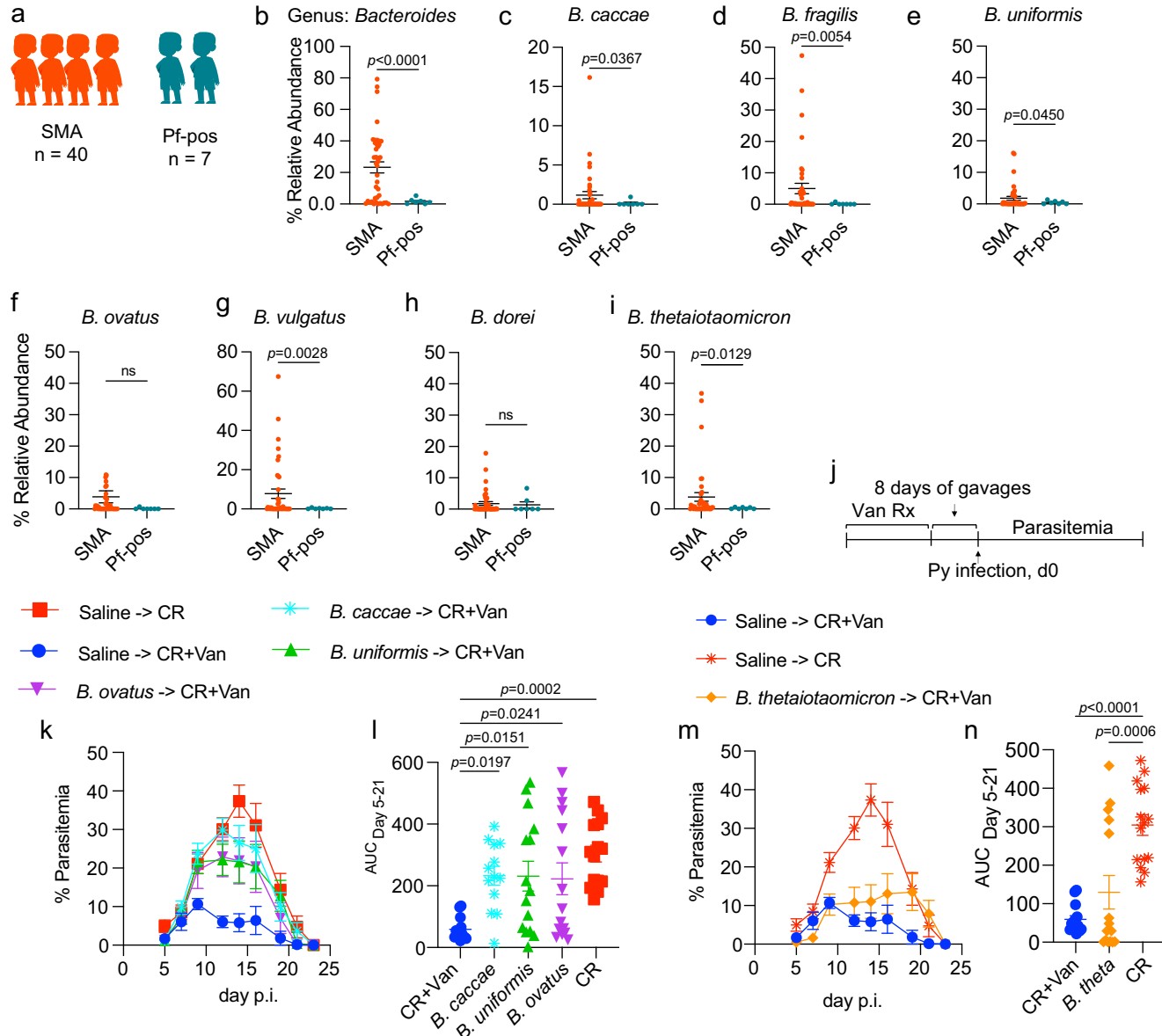

**Fig. 6 | Contribution of human *Bacteroides* in severe malaria. a** Detailed information on the study design and cohort were published earlier[11]. Stool microbiota of kids with severe malaria anemia (*n* = 40) and asymptomatic Pf-pos kids (*n* = 7) were analyzed for relative abundance of *Bacteroides*. **b** Relative abundance of *Bacteroides* genus in Ugandan kids. Relative abundance of *B. caccae* (**c**), *B. fragilis* (**d**), *B. uniformis* (**e**), *B. ovatus* (**f**), *B. vulgatus* (**g**), *B. dorei* (**h**), *B. thetaiotaomicron* (B. theta) (**i**) in Ugandan kids. **j** C57BL/6 mice from CR were treated with 2 weeks of oral vancomycin followed by 8 consecutive bacterial gavages daily. **k** Parasitemia curve of mice gavaged with human strain of *Bacteroides* linked to susceptibility to Py hyperparasitemia. **l** AUC curve analysis. **m** Parasitemia curve of mice gavaged with human strain *B. thetaiotaomicron* linked to resistance to Py hyperparasitemia. **n** AUC curve analysis. All data are mean ± SE (standard error). **b–i** Statistical analysis were performed using unpaired t test with Welch's correction without assuming equal standard deviation with two tailed *p* value. **l, n** AUC were analyzed using one-way ANOVA. Data are merged from two independent experiments with *n* = 5/group in each experiment (**j–n**). Source data are provided in the Source Data file.

*Enterococcus*, *Streptococcus*, and *Ruminococcus* and negatively with *Clostridium* species (Fig. 7g). *Clostridium* species, in turn, can be regulated by *Parabacteroides*, *Odoribacter*, and *Alistipes* (Fig. 7g). These results support that a consortium of bacteria interacting within the broader gut microbiota ecosystem cause susceptibility to Py hyperparasitemia.

Further supporting the contribution of *Bacteroides* species to Py hyperparasitemia, *Bacteroides* related pathways including sphingolipid metabolism pathways and short chain fatty acids pathways and metabolites like N-acetylsphingosine, N-palmitoyl sphinganine, N-palmitoyl sphingosine[30], p-cresol sulphate[31], 2-(4-hydroxyphenyl) propionate[32], and 4-Ethylphenyl sulfate (4EPS)[33] are significantly abundant in the ceca of hyperparasitemia susceptible mice compared to resistant mice (S Fig. 3F and S Fig. 11A–F). Additionally, we have previously shown Py hyperparasitemia-susceptible CR and Env mice have increased abundance of stool propionate compared to Py hyperparasitemia-resistant Tac and Jax mice[34], and *Bacteroides* are known to be among the primary produces of propionate[35,36].

## Discussion

These data demonstrate that members of the *Bacteroides* genus are critical members of a consortium of bacteria that cause susceptibility to severe malaria. As gut microbiome-malaria research is in its infancy and there are notable limitations to the culturomics-based studies that were employed in this study, it is possible that additional gut bacteria (culturable or unculturable) are also critical members of the consortium of bacteria that cause susceptibility to severe malaria. *Bacteroides* is a major and predominant genus of gut bacteria where both

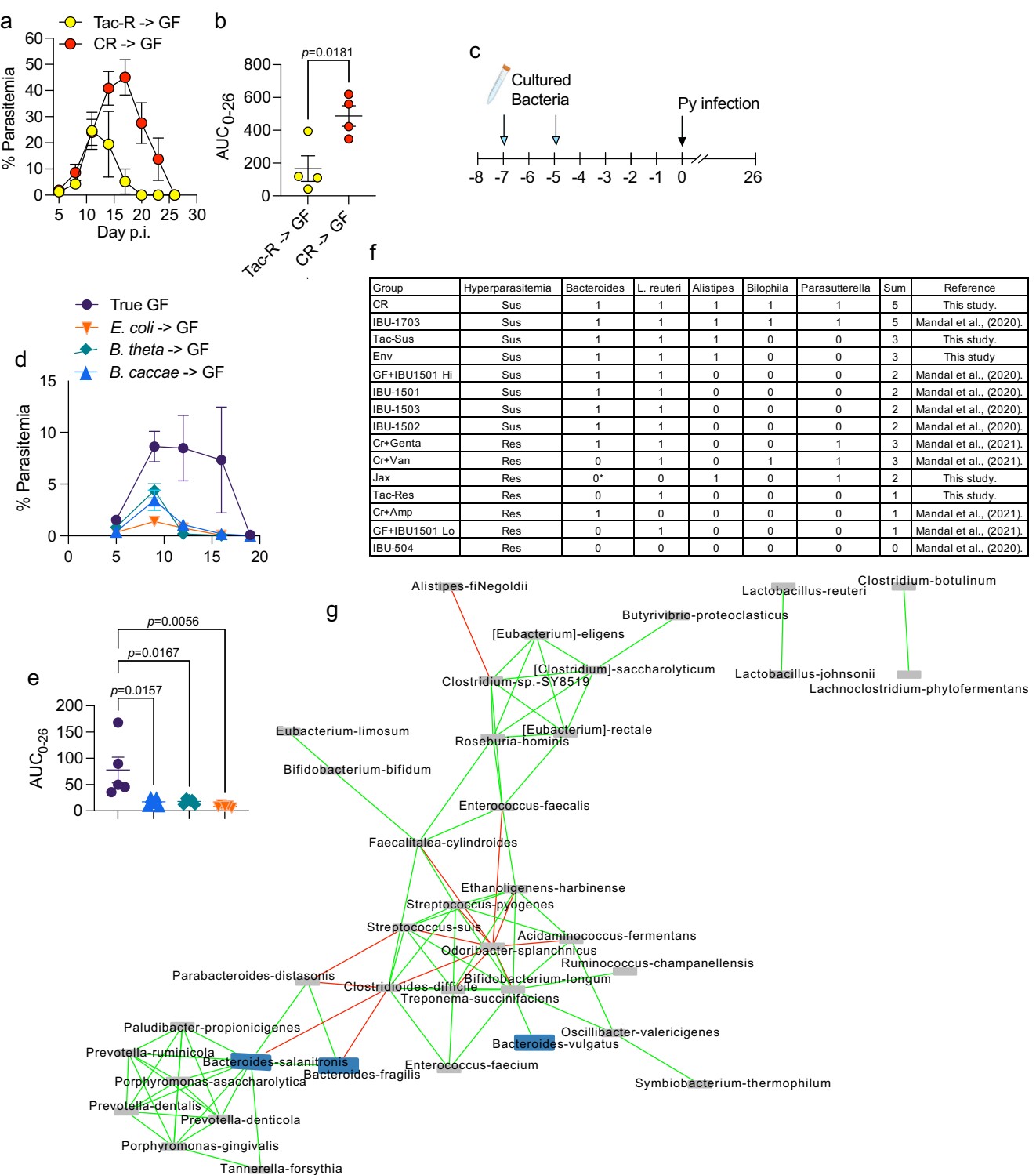

**Fig. 7 | Monocolonization with single *Bacteroides* species does not cause susceptibility to Py hyperparasitemia. a** Parasitemia curve of C57BL/6 germ-free mice transplanted with cecal microbiota from CR and Tac-R mice maintained using gnotobiotic husbandry. **b** AUC analysis. **c** C57BL/6 germ-free mice were gavaged with bacteria on indicated days and rested for 5 days prior to Py infection. **d** Parasitemia curve of germ-free mice treated with *B. caccae*, *E. coli* and *B. thetaiotaomicron*. **e** AUC analysis. Each group contained *n* = 4 mice/group (**a**–**e**). **f** A Consortium of plausible bacteria interacting within the gut to cause susceptibility of Py hyperparasitemia. First column indicates group of mice, and second column shows the Py infection phenotype that are either resistant or susceptible to Py hyperparasitemia. Group names are identical to the previous reports Mandal et al.,

(2020), *BMC Biology* and Mandal et al., 2021, *Cell Reports*. 1: detected; 0: undetected; 0*: Just *B. thetaiotaomicron*. **g** Statistically significant bacterial interaction network within the gut of naïve CR mice. Nodes are bacterial species. Nodes in blue color are *Bacteroides* species. Green edge indicates positive correlation while red edge indicates negative correlation. Cluster indicates co-abundant species. Bacterial interaction network was constructed by CoNet app using Cytoscape. All data are mean ± SE (standard error). Parasitemia were measured on indicated days. AUC were analyzed using one-way ANOVA. **a** Representative of two independent experiments. **d**, **e** Data are from one experiment. Pearson correlation coefficient (r) was set to greater than |0.85| with *p* < 0.05 for bacterial interaction network analysis.

the host and bacteria benefit from a mutualistic relationship[35,37]. *Bacteroides* spp. provide nutrition, amino acids, vitamins, and short-chain fatty acids (SCFAs) to the host and supports other resident intestinal microbiota[35,38]. Additionally, *Bacteroides* species play a role in the stimulation, development, and homeostasis of the immune system and the prevention of bacterial and viral infections[35,39,40]. However, the full pathogenic potential of gut *Bacteroides* spp. is yet to be discovered. *Bacteroides* spp. are sometimes harmful within the gastrointestinal tract and are primarily pathogenic in extraintestinal locations. *Bacteroides* spp. have been isolated from meningitis and brain abscesses; oral infections and abscesses in the neck and lung; associated with Crohn's disease, intra-abdominal abscesses, appendicitis, inflammatory bowel disease, colorectal cancer; and potentially linked to autoimmune disease as well as breast cancer[35,41–43]. This report shows another pathogenic causal link between gut *Bacteroides* and an extraintestinal blood-borne parasitic disease, malaria.

All human and mouse strains of *Bacteroides* spp., except *B. thetaiotaomicron*, tested in this study were able to cause severe malaria as measured by susceptibility to hyperparasitemia. These observations suggest a metabolic function or structural components of *B. fragilis*, *B. caccae*, *B. uniformis*, and *B. ovatus* are distinct from *B. thetaiotaomicron*. This is supported by phylogenetic analysis that demonstrates *B. thetaiotaomicron* is distantly related to the rest of *Bacteroides* based on the full-length 16S rRNA gene (S Fig. 10F).

The exact mechanism of pathogenesis of *Bacteroides* in severe malaria is yet to be deciphered. One potential mechanism by which *Bacteroides* modulate Py hyperparasitemia could be via immune modulation. Mounting evidence demonstrates that gut microbiota modulation of the gut immune system plays a significant role in the development and training of the systemic immune system and influences the outcome of extragastrointestinal diseases like influenza, HIV, diabetes, multiple sclerosis, and autoimmune arthritis, among others[44–46]. For example, segmented filamentous bacteria induce the differentiation of Peyer's patch CD4 T cells into follicular T helper cells, which enter circulation and regulate germinal center formation in auto-antibody production in systemic lymph nodes[45].

Gut *Bacteroides* may impact local intestinal and systemic immunity by producing bioactive molecules. The phylum Bacteroidota and some Proteobacteria have the unique ability to produce sphingolipids, while eukaryotes can ubiquitously make and use sphingolipids as structural membrane components and signaling molecules[47,48]. Sphingolipids, such as sphingosine-1-phosphate, are critical regulators of leukocyte migration through tissues[49]. Germ-free mice colonized with sphingolipid-deficient *B. thetaiotaomicron* resulted in intestinal inflammation[50]. Yet, our data demonstrate colonization of mice with wild-type, sphingolipid-sufficient *B. thetaiotaomicron*[47] does not cause susceptibility to Py hyperparasitemia, which highlights the complexity of these interactions. Further supporting the immune modulatory activity of *Bacteroides*, lower levels of gut *Bacteroides* are associated with inflammatory bowel disease (IBD) like Crohn's disease and ulcerative colitis[51]. *B. fragilis* suppresses intestinal inflammation and regulates intestinal homeostasis via the production of SCFAs[52]. Another immunomodulatory molecule associated with *Bacteroides* is capsular polysaccharides (CPS), which are tightly regulated, dynamic, and firmly attached glycan to the cell surface of *Bacteroides*. CPS are among the most diverse molecules in living systems[53–55]. The structural and physical properties of CPS determine its effect on host health[56]. For example, *B. fragilis* capsular polysaccharide-A (PSA), an immunomodulatory factor, is protective against colitis, encephalomyelitis, colorectal cancer, pulmonary inflammation, and asthma[57]. The ability of several *Bacteroides* spp., except *B. thetaiotaomicron*, to exacerbate severe malaria suggests the malleable expression, diversity (antigenic), and immunomodulatory effect of CPS make them an appealing target for follow-up studies. These observations support the possibility that gut *Bacteroides* could modulate local gut immunity resulting in

alterations in the systemic immune response to *Plasmodium*. Alternatively, *Bacteroides*-derived molecules could directly affect systemic immunity, thereby altering the splenic immune response to *Plasmodium*.

Curiously, higher Py parasitemia was seen in CR+Van mice when gavaged with *Bacteroides* spp. as opposed to Tac-R and Jax mice. This might be due to discrete baseline gut microbiota composition and bacterial interactions within the gut among naïve CR, CR+Van, Tac-R, and Jax mice, reinforcing the hypothesis that *Bacteroides* spp. function within a microbial consortium to cause susceptibility to severe malaria. Pathogenicity of *Bacteroides* spp. do not satisfy Koch's postulates to establish a causal relationship between gut *Bacteroides* and susceptibility to severe malaria. First, CR mice treated with ampicillin and gentamicin have high abundance of *Bacteroides* species[11]. Second, peak parasitemia in Tac-R and Jax mice is ~2- to 4-fold lower when gavaged with a collection of *Bacteroides* (CrBHI) than untreated control CR mice. Third, Tac-R mice treated with *B. fragilis* are not susceptible to Py hyperparasitemia. Fourth, germ-free mice monocolonized with *B. caccae* are not susceptible to Py hyperparasitemia. Thus, the pathogenic potential of these *Bacteroides* is context-dependent on the environment, host conditions, and microbe-microbe interactions.

Beyond *Bacteroides*, other bacteria detected in the CrBHI harvest are also known to be immune modulatory. *Klebsiella oxytoca* is known to cause pathogenesis in antibiotic-associated hemorrhagic colitis (AAHC). Tilivalline is a cytotoxin produced by *K. oxytoca* that is involved with signs of bloody diarrhea and abdominal cramps in humans[58]. *Enterococcus faecalis* and *E. faecium* were detected in Py hyperparasitemia-susceptible mice. In humans, these bacteria are primarily associated with nosocomial infections causing intra-abdominal infections and urinary tract infections (UTI), including others. *E. faecalis* can produce toxins that contribute to the pathogenicity of UTI and gut inflammation[59].

Host immunity is crucial to inhibit and clear blood-stage *Plasmodium* infection[11], specifically splenic germinal center reactions[60,61]. Previously, we have shown that gut microbiota composition impacts the severity of malaria via dynamic modulation of spleen germinal center reactions[11]. Analysis of the host transcriptome revealed differences in immune modulation signatures in circulating PBMCs where B cell receptor signaling and ICOS-ICOSL signaling in T helper cells are upregulated in Py hyperparasitemia-resistant mice compared to hyperparasitemia-susceptible mice, both of which are involved in the formation and maintenance of germinal center reactions[62]. Oral administration of *B. thetaiotaomicron* inhibited the development of allergic airway disease in mice by inhibiting activation of Tregs and inhibition of Th2 response without promoting a Th1 response, possibly by changing the circulating concentration SCFAs[63]. The absence of Th2 cytokines like IL-4 leads to CD4 + T cell help to B cells that control chronic *Plasmodium* infection[64]. *Lactobacillus reuteri*, abundant in Py hyperparasitemia susceptible mice, can promote splenic regulatory T cell development and function and reduce the production of proinflammatory cytokines[65,66]. Splenic regulatory T cells impede acute and long-term immunity to blood-stage malaria by interfering with the formation of germinal centers[67]. Interestingly, L-tryptophan biosynthesis (S Fig. 2B) and tryptophan metabolism (S Fig. 3F) was increased in Py hyperparasitemia resistant mice which can influence Tregs[68]. Farinella et al. (2023) reported significant changes in tryptophan metabolism in the gut at the peak of *P. cynomolgi* infection in rhesus macaque[69]. Likewise, the HNF3A pathway, a FOXA1 transcription factor network, was upregulated in Py susceptible CR mice compared to resistant Tac-R mice. He et al. have shown that FOXA1 overexpression suppressed interferon signaling (IFN) and host immune response to cancer immune response in mice and prostate cancer and breast cancer patients[70]. IFN-γ secretion from innate and adaptive immune cells contributes to immunity against blood- and liver-stage *Plasmodium* infection[71]. Cumulative evidence demonstrates

that gut microbiota modulate IFN responses[72]. Additional research is necessary to delineate the mechanisms by which gut microbiota specifically modulate anti-*Plasmodium* host immunity.

Another exciting finding from our study is the link between gut microbiome-heme metabolism and malaria. Hyperparasitemia susceptible mice gut microbiota (CR and Env) had a higher prevalence of microbial pathways; alpha-Hemolysin/cyclolysin transport system and heme biosynthesis I pathway compared to resistant mice (Tac-R and Jax). Yet, manipulating heme metabolism pathways did not alter Py burden in susceptible (CR) and resistant (Tac-R) mice. Conversely, Dalko et al. reported decreased *P. chabaudi adami* parasitemia in hemin pre-treated mice[73]. Although the heme-biosynthetic pathway of *Plasmodium* is dispensable for the asexual blood-stage, inhibiting heme synthesis by griseofulvin, an antifungal drug, prevented cerebral malaria in mice[74]. The capacity of the host, *Plasmodium* parasite, and gut bacteria to synthesize and regulate heme biosynthesis may have resulted in minimum effect on Py burden. However, the ability of *S. aureus* and other alpha-hemolysin-producing bacteria to exacerbate severe malaria requires further attention. This is particularly relevant as we noted differential hyperparasitemia outcomes when different groups of mice resistant or susceptible to Py hyperparasitemia were treated with an individual (*B. fragilis*) or consortium (CrBHI) of bacteria.

*Bifidobacterium* is associated with decreased *Plasmodium* parasitemia[6,18]. *Bifidobacterium* species are prevalent probiotic bacteria that confer myriad health benefits which have a role in anti-infection, anti-cancer, anti-inflammation, anti-obesity, and others[75]. However, *Bifidobacterium* treatment could not decrease parasite burden in hyperparasitemia-susceptible mice negating the antimalarial effect. Thus, alternative approaches other than conventional probiotics are required to modulate gut microbiota to reduce the severity of malaria.

In conclusion, gut *Bacteroides* spp., except *B. thetaiotaomicron*, was a common denominator amongst the hyperparasitemia-susceptible mouse groups we have investigated. Importantly, unraveling the *Bacteroides* components/products and their interaction with other members of gut microbiota will delineate the mechanistic pathways linking these bacteria to host immunity and control of *Plasmodium* parasite burden. These results will allow the development of gut microbiome-based biologics that fine-tune the abundance of gut *Bacteroides* to prevent severe malaria and associated deaths.

## Methods

### Ethics Statement

Animal procedures and experiments were approved by Institutional Animal Care and Use Committees (IACUC) from the University of Louisville and Indiana University. Written informed consent was obtained from the parents or legal guardians of all study participants. Ethical approval was granted by the Institutional Review Boards at Makerere University School of 1309M42501, date approved: September 23, 2013) and subsequently moved to Indiana University (Ref: 1412213778, date approved: January 20, 2015). The Uganda National Council for Science and Technology approved the study (Ref: HS1522, date approved: May 12, 2013).

### Mice

Six-week-old female C57BL/6 mice were purchased from four different vendors: Charles River Laboratories (CR), Taconic Biosciences (Tac), Jackson Laboratories (Jax), and Envigo (Env). Mice were obtained from CR Isolated Barrier Unit (IBU) R01; Py hyperparasitemia-susceptible Tac mice (Tac-S) from IBU 001501 C and Py hyperparasitemia-resistant Tac mice (Tac-R) from IBU 050401 C; Env mice from IBU 202 A; and Jax from IBU JAXEast:AX4. Mice were housed in a specific pathogen-free (SPF) facility, kept on irradiated NIH-31 Modified Open Formula Mouse/Rat diet (#7913; Envigo, Indianapolis, IN), and provided non-

acidified autoclaved reverse osmosis water ad libitum. Mice were housed in a 12-hour light (6 AM – 6 PM) and 12-hour dark (6 PM – 6 AM) cycle at ambient air temperature (about 22 °C). Mice were acclimatized for 1 week before treatments.

### *Plasmodium* infection and parasitemia analysis

Adult C57BL/6 donor female mice from CR laboratories were infected with thawed *Plasmodium yoelii* 17XNL (Py) infected red blood cell (iRBC) stabilite stored in liquid nitrogen. Py stabilite were resuspended in 1 ml saline to a concentration of approximately $10^5$ iRBCs per 200 µl. Donor mice were intravenously (i.v.) injected in the tail vein with 200 µl Py iRBC suspension. Six days post-infection, 70-80 µl of blood was collected retro-orbitally, and the number of iRBCs was adjusted to $10^5$/200 µl in 0.9% sterile saline (Teknova, Hollister, CA). Experimental mice were injected with $10^5$ Py iRBCs via the tail vein.

Parasitemia was counted using Attune Next (Invitrogen) and BD LSRFortessa (BD Bioscience, San Jose, CA) flow cytometers. About 5 µl whole blood was collected from a tail snip in 100 µl cold 1X PBS. Whole blood was fixed in 0.00625% glutaraldehyde, stained with conjugated antibodies, and ran immediately on a flow cytometer. The staining panel included CD45.2-APC (1:200 dilution) clone 104 (Biolegend, San Diego, CA), Ter119-APC/Cy7 (1:400 dilution) clone TER-119 (Biolegend, San Diego, CA), dihydroethidium (5 mg/ml stock; 1:500 dilution) (MilliporeSigma, St. Louis, MO), and Hoechst 33342 (1 mg/ml stock; 1:1000 dilution) for BD LSRFortessa (MilliporeSigma, St. Louis, MO) and Hoechst 34580 (1 mg/ml stock; 1:1000 dilution) for Attune Next (MilliporeSigma, St. Louis, MO) with. Forward and side scatter singlets were gated on Ter119$^+$CD45.2$^-$ for RBC. For infected RBCs, RBCs were gated on Hoechst$^+$Dihydroethidium$^+$ to find the number of RBCs containing Py parasite DNA and RNA.

### Ceca sampling for shotgun metagenomics

Mice were anesthetized using isoflurane (Pivetal) and cervically dislocated inside a biosafety cabinet. Mice were doused with 70% ethanol and opened using sterile scissors and forceps (Ambler Surgical). Ceca were removed and squeezed into 2 ml cryogenic tubes (Corning). Ceca was longitudinally cut open and laid on a flat surface. A sterile micro glass slide (VWR) was used to hold at one end of the ceca, and with another glass slide, the mucosal surface was scrapped off and collected in the same cryogenic tube. Cryogenic sample tubes were immediately placed in liquid nitrogen and stored at −80 °C. For DNA extraction, samples were thawed on ice and vortexed vigorously for 10 min with a stainless-steel bead of 5 mm (Qiagen). About 200 µl samples were used for DNA extraction, as described below.

### Bacterial strains and growth conditions

Bacteria were either isolated from the ceca of CR mice or purchased from the American Type Culture Collection (ATCC). Briefly, mice were euthanized, ceca content were pooled, serially diluted in 1X PBS, plated on agar plates, and incubated. All the bacteria were grown anaerobically (5 % $CO_2$, 5% $H_2$, and 90 % $N_2$) for 48 hours at 37 °C. Single colonies were picked and streaked again on an agar plate to obtain pure colonies and stored in 50% glycerol stock at −80 °C. Human isolates of *Bacteroides thetaiotaomicron* (ATCC 29148), *B. uniformis* (ATCC 8492), *B. ovatus* (ATCC 8483), and *B. caccae* (ATCC 43185) were obtained from ATCC and glycerol stocks were made following manufacturer protocol. Mouse isolate *B. fragilis* was isolated using a selective medium Bacteroides Bile Esculin Agar (BBE) plate (Fisher Scientific). Mouse isolate *Escherichia coli* was isolated from Brain Heart Infusion (BHI) agar (VWR). All the *Bacteroides* species were cultured in Wilkins-Chalgren Anaerobe (Wilkins) broth (Thermo Fisher Scientific) and *E. coli* on BHI medium. Bacteria were identified using 16S rRNA full-length sequencing (Eurofins Genomics, Louisville) and MALDI-TOF (Indiana University, School of Medicine, Indianapolis). *Micromonospora auranticia* (ATCC 27029) was purchased and grown following the ATCC

protocol. Briefly, *M. auranticia* was incubated in Yeast malt broth (Sigma-Aldrich) on shaking rack at 37 °C for 5 days aerobically and homogenously mixed using sonicator before treating mice.

## Culturomics and gavage regimen

Naïve Py hyperparasitemia susceptible 6-week-old C57BL/6 CR mice were rested for 1 week after delivery. Ceca content from five mice were pooled in 5 ml sterile saline inside a biosafety cabinet. Ceca contents were vigorously vortexed to homogeneously mix the content. Ceca contents were serially diluted 10-fold up to six dilutions in 1 ml saline and plated on respective agar plates. Dilutions were spread on an agar plate using sterile glass beads (Millipore Sigma). Plates were cultured anaerobically for 48 hours at 37 °C. Bacterial growths were harvested from each of the six dilutions agar plates in the respective broth medium using hockey shaped sterile inoculating loop (Fischer Scientific), centrifuged at 4400 rpm for 10 mins, resuspended in a small volume of respective medium, and 50% glycerol stocks were made and stored at −80 °C.

Glycerol stocks were inoculated in 10 ml of the respective media using inoculating loop under a flame. After 48 hours of incubation in the above anaerobic conditions, the bacterial pellet was obtained with centrifugation at 4400 rpm for 10 mins, resuspended in 1 ml sterile saline. 200 µl resuspended bacteria were gavaged every day from approximately 7 days before Py infection to 7 days after Py infection. Variations in the number of gavages are described in the figure legends. Gavage treatments were done with reusable 20-gauge gavage needles containing a 2.25 mm ball diameter and 3.81 cm length (Cadence Science, Place).

## Bacterial growth media and antibiotics

All bacterial media used in this study were commercially purchased. Luria Broth (LB) Base (Invitrogen) De Man, Rogosa and Sharpe agar (MRS, Oxoid), Reinforced Clostridial (RC) Medium (BD Difco), Tryptic Soya Agar (TSA, BD Diagnostic), Wilikins-Chalgren Agar (Sigma Aldrich) and broth (Oxoid), Brain Heart Infusion Agar (Criterion), Blood Agar Plates with 5% Sheep Blood in TSA Base (Hardy Diagnostics), Selective Streptococcus Agar (BD Diagnostic), and Aureus ChromoSelect Agar Base (Millipore, Place) were used for culturomics. Yeast malt broth (Sigma-Aldrich) was used to grow *M. auranticia*. Oral vancomycin (VWR) was given in drinking water for 1 to 2 weeks at 100 mg/400 ml water and changed weekly.

## Blood collection for transcriptomics

C57BL/6 mice from CR were treated with vancomycin (CR+Van) in drinking water for 1 week. CR, CR+Van, and Tac-R mice were infected with Py. Mice were anesthetized with isoflurane using an animal anesthesia machine (Euthanex). Immediately 100 µl of blood was collected retro-orbitally using a plain Natelson Capillary tube (DWK Life Sciences) and mixed in an equal volume of RNAlater (ThermoFisher Scientific), mixed gently, and stored at −80 °C until RNA extraction. Whole blood was collected from CR, CR+Van, and Tac mice on 0-, 5-, and 10-days post-infection.

## DNA extraction and sequencing

DNA was extracted from whole ceca content, fecal pellet, and culturable bacteria using QIAamp PowerFecal DNA kits (Qiagen). DNA was measured using Qubit 3 Fluorometer (Invitrogen). DNA library preparation and sequencing were performed at UC Davis for ceca shotgun metagenomics, Washington University for 16S rRNA gene amplicon-based sequencing, and IU Medical genomics core for shotgun metagenomics of bacterial culture. Shotgun metagenomics was sequenced using HiSeq 4000 (ceca content) or NovaSeq 6000 (bacterial culture) with 150 bp paired-end sequencing. Fecal pellet DNA was sequenced using Multiple 16S Variable Region Species-level Identification (MVRSION) approach using HiSeq 4000[76].

## RNA extraction and sequencing

Blood stored in RNA later at −80 °C was thawed on ice. Total RNA was extracted using RNeasy Mini Kit (Qiagen) following the manufacturer's protocol. Ribosomal RNA was removed using the NEBNext rRNA Depletion kit (New England Biolabs) and globin mRNA was removed using a GLOBINclear kit (Thermo Fisher Scientific). The purity and concentration of RNA were quantified using a NanoDrop (Thermo Scientific) microvolume spectrophotometer. Library preparation and sequencing were performed at the IU Center for Medical Genomics Core. PolyA mRNA sequencing was performed using NovaSeq 6000 with 100 bp PE sequencing.

## Shotgun metagenome analysis

Microbiota profiling using shotgun metagenomics was performed using multiple bioinformatics pipelines. Trimmomatic[77] was used to filter poor-quality reads and trim poor-quality bases and adapter sequences from raw reads. Murine host reads were filtered from microbial reads using bowtie2 and samtools (https://www.metagenomics.wiki/tools/short-read/remove-host-sequences)[78,79]. Quality microbial forward and reverse reads were used for downstream analysis using MetaPhlan2[80], Clark[81], de novo assembly and binning, and CZID[82]. MetaPhal2 used 1 M unique clade-specific marker genes (mpa_v20_m200_marker database). Reads were classified with CLARK using 31-mer against the complete genome of bacteria and archaea from NCBI/RefSeq. De novo assembly and binning were performed following the DIBSI Metagenomics workshop pipeline (https://2017-dibsi-metagenomics.readthedocs.io/en/latest/index.html). Taxonomic classification of bins were conducted using PhyloSift[83]. Microbial reads were uploaded to the CZID server for taxonomic classification. Biom table from combined sample taxon results using nucleotide (NT) database of NCBI with total reads were downloaded. Taxonomic classification from various pipelines was modified to be imported to QIIME2[84] for visualization, core diversity analysis, and visualization.

Microbial pathway analysis was performed using HUMAnN2[85] and GhostKoala[19]. UniRef90 database was used to accurately profile microbial pathway abundance using HUMAN2. The detail of the GhostKoala pipeline is shown in S Fig. 2C. Briefly, Prokka[86] was used to determine gene content ( ~ 2.8 million genes) in the contigs generated by MEGAHIT[87]. Salmon[88] was used to quantify the gene content in each sample from a database constructed using ~2.8 million genes. Core genes shared by at least 50% of mice in each group were made. The amino acid sequence of these genes was uploaded consecutively to GhostKoala, and pathway analysis files were downloaded. Module files were parsed to quantify the number of genes hit to a module. Log 2-fold change was calculated between Py hyperparasitemia susceptible and resistant groups. Bacterial interaction network was constructed from naive CR mice by CoNet App[89] using Cytoscape 3.3[90].

## 16S rRNA gene analysis

Amplicon-based microbiota profiling was accomplished by the Multiple 16S Variable Region Species-level Identification (MVRSION) approach, which can sequence all the hypervariable regions of the 16S rRNA gene with 12 primer pairs[76]. MVRSION was performed at Genome Technology Access Center (GTAC), Washington University, Saint Louis. The observed taxonomic unit (OTU) table at the species level was constructed by GTAC. OTU table was further analyzed by importing QIIME2 for core-diversity analysis and statistical analysis. The number of reads per sample was rarefied by the sample containing the lowest number of reads as described in figure legends.

## Metabolomics

Cecal contents were collected, and flash frozen in liquid nitrogen. Sera were extracted from whole blood collected using retro-orbital bleed

and kept at −80 °C. Samples were shipped to Metabolon on dry ice. Briefly at Metabolon, samples were maintained at −80 °C until processing. Samples were prepared at Metabolon using the automated MicroLab STAR® system from Hamilton Company along with standards. The controls included extensively characterized human plasma, ultra-pure water, and solvent used for extraction to assess instrument performance and aid chromatographic alignment. Proteins were removed by vigorous shaking for 2 min (Glen Mills GenoGrinder 2000) followed by centrifugation. Samples were analyzed on positive and negative ion mode (RP)/UPLC-MS/MS and HILIC/UPLC-MS/MS with negative ion mode ESI. Orbitrap mass analyzer operated at 35,000 mass resolution. The MS scan range varied slighted between methods but covered 70-1000 m/z. Metabolon performed raw data extraction, peak-identification and QC processing using their hardware and software. Peaks were quantified using the area-under-the-curve. Data were normalized by the medians equal to one and normalizing each data proportionately. Additional normalization was done in metabolite levels to account for differences in the amount of starting material. Metabolon prepared and provided the metabolite and pathway analysis spreadsheet. Raw spectral data is deposited at https://www.ebi.ac.uk/metabolights/reviewerc2bcbc66-8583-45a3-bbd7-aa85632f07fe.

For data visualization and statistical analysis, the normalized data were imported in MetaboAnlayst 5.0[91]. Data was analyzed as one factor and data filtering were done with interquartile range (IQR) at 25% with variables that are near-constant throughout the experimental conditions. Additional normalizations were not performed. Heatmap were generated using Pearson distance and average clustering method.

## Bulk transcriptomics analysis
Whole blood transcriptome was analyzed using Salmon[88] to produce a highly accurate transcript-level quantification. Briefly, quality control was performed using Trimmomatic[77]. A combined database was made from C57BL/6 mouse using NCBI's GRCm38.p6 transcriptome and PlasmodDB *P. yoelii* 17XNL (version 46) transcriptome to map and quantify transcripts in parallel. Mouse transcriptome included predicted mRNA and non-coding RNA along with known transcripts. *Plasmodium* transcriptome is not shown in this manuscript. Differential gene expression of mouse transcriptome was performed using statistical tests implemented inside Trinity[92]. Canonical pathway analysis was performed using commercially available Ingenuity Pathway Analysis (IPA) software (QIAGEN IPA). Gene set enrichment analysis (GSEA) was done using the Human Molecular Signatures Database (MSigDB) hallmark gene sets converted to mouse orthologous genes[93]. GSEA was also performed using the WEB-based Gene SeT AnaLysis Toolkit using databases like KEGG, Panther, Wikipathway, and Recatome for functional pathway analysis[94].

## Heme measurement and pathway modulation
The level of total heme in the blood was quantified using Heme Assay Kit (Sigma-Aldrich). Briefly, 2 µl of blood was collected using a tail snip and diluted in 100-fold ultrapure water. Heme was measured following manufacturer's recommendations. The impact of heme metabolism on Py parasitemia was modulated using three compounds. Tin protoporphyrin (SnPP, Cayman Chemical Company) was dissolved in 25% dimethyl sulfoxide (DMSO, Thermo Fisher Scientific) and injected intra-peritoneally (IP) at 0.42 mg/mice every other day from day 0 to 13 days post Py infection. Hemopexin (Hx, Millipore Sigma) was dissolved in saline and injected IP at 10 µg/mice every other day till day from 0 to day-8 post-Py infection. Hemin (Sigam-Aldrich) was dissolved in 0.1 M NaOH and injected IP at 0.2 mg/mice every other day from 0 to 10 days post-infection.

## Hemoglobin quantification
The amount of hemoglobin in the blood was measured using Germaine Laboratories AimsStrip Hemoglobin test system and AimStrip Hemoglobin (Germaine Laboratories) following the user manual. The Hemoglobin test system was set up and automatically calibrated using a code chip. Approximately 10 µl tail snip blood was applied to the center hole of the specimen area of the test cartridge. The hemoglobin value was displayed within 15 seconds.

## Human participants
A detailed description of the participants was published previously[11]. Children aged 0.5–4 years from Kampala and Jinja in Uganda were enrolled as part of a clinical observation study. Stool samples were collected during and after hospitalization from children with severe malaria anemia (SMA, $n = 40$) and at enrollment from community children with asymptomatic *Plasmodium falciparum* infection (Pf-pos, $n = 7$) and frozen immediately at −80 °C. Stool samples were kept frozen until further processing for DNA extraction. SMA was defined as *P. falciparum* smear or RDT positive and serum hemoglobin level ≤ 5 g/dL. Pf-pos was defined as community children that were *P. falciparum* microscopy positive. The abundance of gut *Bacteroides* in the stool of children with SMA and Pf-pos was reanalyzed from formerly sequenced data[11].

## Gnotobiotic husbandry
Female germ-free C57BL/6 mice, 6-week old were acquired from Charles River Laboratories and housed in autoclaved Techniplast ISOCage P cages (Techniplast) with ALPHA-dri bedding (Shepherd Specialtiy Papers, Inc) and Bed-r'Nest nesting material (The Andersons Plant Nutrient Group). Light cycle, feed, and autoclaved water were identical to SPF mice. Germ-free mice rested for 24 hours, followed by bacterial gavage twice on alternate days, and rested for additional five days before Py infection. Parasitemia was tracked as described earlier. Germ-free status was assessed every week by shipping fecal pellets to IDEXX BioAnalystic. Sterility was assessed using universal 16S rRNA sequencing and culturing for fungal and bacterial growth.

## Statistical analysis
Statistical testing was done using GraphPad Prism 9 software, QIIME2, Trinity, and R packages. Specific statistical tests and significance cut-offs are described in figure legends. Group comparisons were considered significantly different at P < 0.05. The area under curve (AUC) was analyzed using the trapezoidal rule as performed previously[11].

## Reporting summary
Further information on research design is available in the Nature Portfolio Reporting Summary linked to this article.

## Data availability
Source Data are provided with this paper. All raw sequences are deposited to NCBI Sequence Read Archive (SRA) under the following BioProject accession code: PRJNA962898, PRJNA962885, PRJNA962866, PRJNA962119, PRJNA961982. Raw spectral data from metabolomics study is deposited to EMBL-EBI's Metabolights under the study accession code: MTBLS3449. Source data are provided with this paper.

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

## Acknowledgements

We are thankful to Drs. Tuan M. Tran, Martin Richer, Mark Kaplan, Jay C. Vornhagen, and Brittany D. Needham for constructive feedbacks during manuscript preparation. This work was supported by grants from the National Institute of Allergy and Infectious Disease of the National Institutes of Health (NIH) (R01AI123486 and R01AI148525 to N.W.S. and R01NS055349 to C.C.J.) and funds from the University of Louisville and Indiana University School of Medicine (to N.W.S.). Support provided by the Herman B. Wells Center (to N.W.S. and C.C.J.) was in part from the Riley Children's Foundation. The project described was supported by the Indiana University Health-Indiana University School of Medicine Strategic Research Initiative (to N.W.S.). The content is solely the responsibility of the authors and does not necessarily represent the official views of the NIH. We would also like thank Kristin Marie Van Den Ham, Layne Bower, Elizabeth M. Fusco, and Olivia J. Bednarski for help during gnotobiotic experiments.

## Author contributions

Conceptualization, R.K.M. and N.W.S.; formal analysis, R.K.M.; investigation, R.K.M., A.M., and J.E.D.; writing – original draft, R.K.M.; writing – review & editing, R.K.M., N.W.S., C.C.J., R.N., A.M., and J.E.D.; visualization, R.K.M.; funding acquisition, N.W.S.

## Competing interests

The authors declare no competing interests.
