## [Peer Review File · Nature Communications]

Reviewers' expertise:

Reviewer #1. Mouse genetics / infectious diseases / malaria.

Reviewer #2. Plasmodium infection / pathogenesis / microbiome.

Reviewer #3. Microbiome / metagenomics / metatranscriptomics.

Reviewers' comments:

Reviewer #1 (Remarks to the Author):

Here, Mandal et al examine the role of specific gut microbiota implicated in human susceptibility to severe malaria in promoting sensitivity to high parasitemia in mice. Using 5 commercial sources of C57Bl/6 mice with different susceptibility to *Plasmodium yoelii* hyperparasitemia, the authors conducted metagenomics of cecal samples to identify major bacterial species that might be associated with host resistance to infection. Multiple cultures were used to gavage hyperparasitemia-susceptible CR (Charles River) or resistant Tac-R (Taconic) mice but failed to change host susceptibility to developing parasitemia. A second strategy was used to identify culturable *Bacteroides* species from CR mice, and gavaged into either Jackson Laboratories (JAX) mice or CR mice cleared of microbiota through treatment with vancomycin that were now resistant to *P. yoelii* parasitemia. This strategy worked to increase the circulating parasite load, and was used to implicate *Bacteroides* species as a risk factor for severe malaria. Furthermore *Bacteroides* enriched from the stool of children with severe malaria anemia also increased the parasite burden in *P. yoelii*-infected vancomycin-treated CR mice, thus further implicating this bacterial genus in disease severity.

In general the work appears to be important and of high scientific interest. However, the paper is disjointed and reads as though there is a lot of negative data, which may be accurate, or it may be a reflection of the mouse models and whether the microbiota or genetic drift in the colonies is determining the initial outcomes (see comment 1 below).

Major comments:

1) The experimental concept of the paper is that specific microbiota can be linked to parasite burden in the *P. yoelii* mouse model. However, until the first treatment of CR mice with vacomycin in figure 4I, this is not established making the data preceding this difficult to interpret. A role for genetic drift in the various C57bl/6 mouse colonies needs to be ruled out in determining the initial resistant or sensitive phenotypes of hyperparasitemia by treating all strains with van and determining their relative sensitivity. This will determine how valid the initial approaches (Figure 1-4H) are as being able to determine roles for microbiota at all.

2) The work with *S aureus* is incomplete and confusing and does not add to the paper. I recommend removal.

Reviewer #2 (Remarks to the Author):

Overall, this is a very interesting paper that uses a combination of approaches to determine whether particular clades of gut bacteria can cause an increase in severity to malaria. The strengths of the paper are the combination of descriptive and experimental techniques (both studying the microbial composition in human patients, and also performing controlled studies of causality in mice) and the rigor of data analysis (using multiple different analytical approaches and comparing the results across these different methods). I did end up with some level of hesitation on how causative this relationship is between *Bacteroides* and “susceptibility to severe malaria” (as indicated in the title). My hesitation is for two main reasons: (1) the data from the murine studies consistently point to a conglomerate of bacteria in causing severe disease. I understand *Bacteroides* are enriched, but is it really causative? And (2) the human studies are set up to answer a different question – all individuals have malaria and the comparison is being done of those with asymptomatic versus severe. So the authors cannot really conclude from the human data that *Bacteroides* is causative. Could it in fact go the other way (that the severity of infection, e.g. high immune response) causes a disturbance in the gut microbiome in those humans? That is feasible based on what is seen in many other diseases (that poor health is associated with a disruption to the gut microbiome). The paper is very thorough and has a lot of great techniques but I did feel a bit hesitant about the conclusion that *Bacteroides* causes severe malaria, which I believe is the point argued by these authors.

Major points:

Line 62: The ceca was a focus. The authors should explain why they chose the ceca in particular.

Line: 95 (Figure 1F-I), it is not clear how the data was analyzed to conclude that “all five groups had significantly different microbial compositions”. What statistics were used to conclude this? Furthermore, in line 98, it is stated that “broad taxonomic analyses do not distinguish Py hyperparasitemia-resistant and -susceptible mice”. I cannot seem to reconcile these two seemingly conflicting phrases, one stating that all groups were significantly different than one another” and the other stating that they could not be distinguished.

Line 101: Since functional profiling was done of the metagenome, how did the authors determine that they had sufficient read depth to do this analysis and was it consistent across their samples?

Line 106: The authors indicate that there is “distinct clustering of all five groups of mice except for Taconic” but in Figure S Figure 2A, it looks like Env is also overlapping with Tac-S, in addition to Tac-R overlapping with Tac-S as mentioned by the authors. So in fact, it would appear that just 2 groups are actually distinct clusters on their own (Jax and CR). This should be clarified.

Line 154: It would be helpful to have more clarify on why “whole blood bulk metatranscriptomics” was done in this study. From the wording of metatranscriptomics, I assumed this was geared towards studying microbial transcripts in the blood, but I believe that this is focused solely on host transcripts, which would be better termed “whole blood transcriptomics”. That being said though, it still doesn’t quite explain why focusing on the whole blood (and presumably therefore, immune cells) would help to test the hypothesis stated in line 151 (“that Py hyperparasitemia susceptible mice upregulate the heme metabolic pathway due to enriched gut metagenomic potential for alpha-hemolysin/cyclolysin transport systems and increased capacity of gut bacteria for heme biosynthesis pathways”). The connection between the activity of the gut microbes and the immune cell transcripts is unclear and does not appear to test the hypothesis stated.

Line 172: This is an interesting hypothesis raised about bacterial interactions with red blood cells. However, is there variability in the level of bacterial translocation / gut leakiness in these different animals from different vendors? I wonder whether the leakiness factor could be more important than the gut microbial composition if the idea is that the bacteria are lysing red blood cells.

Line 266: The connection to the human studies seems unclear. The idea of the mouse studies focuses on whether healthy mice having certain types of gut microbes impacts their level of parasitemia upon malaria infection. However, in the human study, the gut microbes are studied in children with different levels of disease severity from malaria. Since this is not a longitudinal study in humans that parallels what is done in mice, how do the authors rule out that the effect is not the other way around in humans (the severe malaria disease itself causing gut microbial changes?)

Line 299: The authors study the relatedness of the Bacteriodes species using phylogeny of the 16S gene, but this doesn't really seem sufficient to understand the level of similarity between these species. Are there other ways to look more thoroughly into the differences between B.t. versus the other B. species?

Discussion: It would be helpful for the authors to compare their results with a 2023 publication by Farinella et al (<https://www.frontiersin.org/articles/10.3389/fcimb.2022.1058926/full>) and comment on whether results are similar or different. For example, Farinella et al describes an increase in the genetic capacity for tryptophan biosynthesis in the gut microbes during malaria infection. Figure S2B in this paper seems to indicate a similar finding in terms of significance for the tryptophan biosynthesis pathway, and may be worth commenting on.

Discussion: Was a complete fecal transfer done to test the hypothesis that the complex of bacteria mediates disease severity? This seems like a good way to test that stated hypothesis.

Methods: What is the expected result of the heme assay kit when applied to lysed red blood cells? The authors indicate that they lysed red blood cells in water prior to measuring heme, but was the goal to measure heme inside of RBCs or free heme? This part could be better clarified.

Methods: How were the stool samples collected in the humans? (this is not explained in the paper). Although they cite another publication, it may be helpful to briefly explain here.

Minor

Line 23: P. malariae is misspelled

Line 591: PlasmoDB is misspelled

Line 622: May be useful to any relevant IRB approval numbers to this section.

Reviewer #3 (Remarks to the Author):

This manuscript outlines a multiomics approach to investigate how specific species of *Bacteroides* may be causally linked to the risk of severe malaria. They found that isolates of *Bacteroides caccae*, *Bacteroides uniformis*, and *Bacteroides ovatus* but not *Bacteroides thetaiotaomicron* caused susceptibility to severe malaria in mice. However, the pathogenic potential of gut *Bacteroides* towards susceptibility to severe malaria appears to be dependent on additional gut microbiota, indicating a consortium effect in severe malaria.

In general, this paper is well-written. The overall scientific approach is reasonable albeit lacking in some critical details, as outlined below. The authors appear to be quite knowledgeable in this field and have related previous publications that are relevant. Specific comments are listed below:

1. The manuscript title is over-reaching and is not supported by the content of the manuscript – please revise this to be more accurate.
2. Page 1 – lines 16-18; this statement is very concerning and somewhat undermines the basic tenet of this manuscript. If the “pathogenic potential of gut *Bacteroides* is dependent on the (functions) of additional gut microbiota,” then their approach based on metagenome sequencing is too limited.
3. Much of the text in pages 4 and 5 might be more appropriate in the Method section as opposed to Results.
4. Page 4, line 72 – this sub-header is confusing – what does “enriched genomic potential” mean? This is too vague.
5. The authors correctly point out (page 5, line 98) that “broad taxonomic analyses do not distinguish disease phenotypes,” but they fall into a similar very problematic assumption that “more abundant genes in the genome” suggest function and then link that to a particular pathway. The authors are to be severely cautioned here – there is now copious evidence from numerous publications that predicting actual function from “metagenomic potential” is very dangerous and has led to numerous confounding results. As the authors point out, the actual metabolic activities of the gut microbiota are very context dependent and thus are best measured perhaps by metaproteomics and metabolomics. Why didn't the authors pursue these more informative systems? At present, they metagenome “potential predictions” provide very little substantive activity information and in fact could be highly misleading.

6. Page 7, line 129 – here and elsewhere are beautiful examples of where metabolomics could reveal some critical insights into actual “function” rather than “potential function.” Sad that there was a missed opportunity.

7. Page 8, line 154 – it is confusing what the authors mean by “whole blood bulk metatranscriptomics?” This looks to be mouse transcriptomics – where is the “metatranscriptomics?”

8. Page 9, line 182 – this section is nice but basically shows that some culturable gut bacteria increase parasite burden – how is that related to the complex multi-species function in the actual mouse gut ecosystem? As such, this is an interesting but quite disconnected.

9. Page 11, line 217 – this section is focused on cultured bacteria and may totally miss other bacteria that are casual agents or co-members necessary for infection. These types of “culture studies” are valuable but do not adequately reflect the context-dependent complexity of the actual in-vivo systems. The authors need to be more clear to qualify their approach and findings and acknowledge the limitations. Otherwise, they are finding some bacterial types that can cause hyperparasitemia but this might not translate accurately into what is actually happening/controlled in the in-vivo systems.

10. Page 13, line 273 – the authors are to be cautioned here – linking the abundance of bacteria at the genus level is very broad – there are likely many members within that genus that are highly different in function. Thus, this classification at a quite broad level and implicated by abundance at the “genomic potential function level” has been and is likely to continue to lead to confounding information.

11. Page 14, line 302 – this statement is over-reaching and reflects some basic inherent problems with the experimental approach of this work. To make this statement based primarily on metagenomic potential function and culturable bacteria is too limited – there could be (maybe likely are) other unculturable members that are actually much more active in the disease process. This needs to be accurately acknowledged in the Discussion. Even so, the authors confess that “context matters” in that *Bacteriodes* spp. are “sometimes harmful in the GI tract” – line 310.

12. Page 20, final section – it is not clear what the technical value of this manuscript is? Certainly, some gut *Bacteroides* are common among hyperparasitemia-susceptible mice but some of that information was already known. How this relates to the actual context-dependent actual metabolic

activities in the in-vivo mouse is unclear? This is apparently why the authors refer in numerous places in the manuscript that their data “may imply or suggest...” Even though they tout a multi-omics approach, many of their techniques are too distant from monitoring the actual metabolic activities in the in-vivo system and thus their results seem disconnected from their goals.

We would like to thank the reviewers for the time they invested to the review of this manuscript and the thoughtful comments and suggestions to improve the quality of the report. These critiques have been informative in our effort to improve this manuscript. We have provided a point-by-point response to each of the comments below.

Reviewers' expertise:

Reviewer #1. Mouse genetics / infectious diseases / malaria.

Reviewer #2. Plasmodium infection / pathogenesis / microbiome.

Reviewer #3. Microbiome / metagenomics / metatranscriptomics.

Reviewers' comments:

Reviewer #1 (Remarks to the Author):

Here, Mandal et al examine the role of specific gut microbiota implicated in human susceptibility to severe malaria in promoting sensitivity to high parasitemia in mice. Using 5 commercial sources of C57Bl/6 mice with different susceptibility to Plasmodium yoelii hyperparasitemia, the authors conducted metagenomics of cecal samples to identify major bacterial species that might be associated with host resistance to infection. Multiple cultures were used to gavage hyperparasitemia-susceptible CR (Charles River) or resistant Tac-R (Taconic) mice but failed to change host susceptibility to developing parasitemia. A second strategy was used to identify culturable Bacteroides species from CR mice, and gavaged into either Jackson Laboratories (JAX) mice or CR mice cleared of microbiota through treatment with vancomycin that were now resistant to P. yoelii parasitemia. This strategy worked to increase the circulating parasite load, and was used to implicate Bacteroides species as a risk factor for severe malaria. Furthermore Bacteroides enriched from the stool of children with severe malaria anemia also increased the parasite burden in P. yoelii-infected vancomycin-treated CR mice, thus further implicating this bacterial genus in disease severity.

In general the work appears to be important and of high scientific interest. However, the paper is disjointed and reads as though there is a lot of negative data, which may be accurate, or it may be a reflection of the mouse models and whether the microbiota or genetic drift in the colonies is determining the initial outcomes (see comment 1 below).

Major comments:

1) The experimental concept of the paper is that specific microbiota can be linked to parasite burden in the P. yoelii mouse model. However, until the first treatment of CR mice with vacomycin in figure 4I, this is not established making the data preceding this difficult to interpret. A role for genetic drift in the various C57bl/6 mouse colonies needs to be ruled out in

determining the initial resistant or sensitive phenotypes of hyperparasitemia by treating all strains with van and determining their relative sensitivity. This will determine how valid the initial approaches (Figure 1-4H) are as being able to determine roles for microbiota at all. The reviewer raises an important point. This has been clarified in the revised manuscript by noting in the Introduction (lines 55-57) and at the beginning of the Results (lines 77-83) that our previous publications have established that gut microbiota in mice from the different vendors used in the study have been shown to be a causal risk factor for susceptibility to severe malaria. Within the referenced publications we have shown that transfer of ceca contents from mice obtained at different vendors with susceptibility or resistance to Py hyperparasitemia into genetically identical germ-free mice results in susceptibility or resistance, respectively, following Py infection. Transfer of ceca contents into germ-free mice and recapitulation of the susceptible and resistant phenotypes also eliminated genetic drift between the various C57BL/6 mouse colonies as a contributing factor. These revisions establish at the onset of the Results that gut microbiota are different between these vendors, they cause differential susceptibility to Py, and exclude genetic drift as a contributing factor.

2) The work with S aureus is incomplete and confusing and does not add to the paper. I recommend removal.

We appreciate the perspective of the reviewer and comment regarding the lack of clarity of these experiments. We have revised the rationale (line 168-174) to explain why these studies were pursued. Our results demonstrate that treatment of Tac-R mice with S. aureus increases susceptibility to Py hyperparasitemia, albeit minimally. Whereas the results do not show a profound effect of S. aureus treatment on Tac-R mice, it might be that additional bacteria absent in Tac-R mice are required for S. aureus to exert a more profound effect, similar to what we've reported in this manuscript regarding Bacteroides. Collectively, given the results support a potential role we have included the results as they provide support for further investigation.

Reviewer #2 (Remarks to the Author):

Overall, this is a very interesting paper that uses a combination of approaches to determine whether particular clades of gut bacteria can cause an increase in severity to malaria. The strengths of the paper are the combination of descriptive and experimental techniques (both studying the microbial composition in human patients, and also performing controlled studies of causality in mice) and the rigor of data analysis (using multiple different analytical approaches and comparing the results across these different methods). I did end up with some level of hesitation on how causative this relationship is between Bacteroides and "susceptibility to severe malaria" (as indicated in the title). My hesitation is for two main reasons: (1) the data from the murine studies consistently point to a conglomerate of bacteria in causing severe disease. I understand Bacteroides are enriched, but is it really causative? And (2) the human studies are set up to answer a different question – all individuals have malaria and the comparison is being done of those with asymptomatic versus severe. So the authors cannot really conclude from the human data that Bacteroides is causative. Could it in fact go the other way (that the severity of infection, e.g. high immune response) causes a disturbance in the gut

microbiome in those humans? That is feasible based on what is seen in many other diseases (that poor health is associated with a disruption to the gut microbiome). The paper is very thorough and has a lot of great techniques but I did feel a bit hesitant about the conclusion that *Bacteroides* causes severe malaria, which I believe is the point argued by these authors.

Regarding hesitation #:

- 1) We fully agree our data point towards a “conglomerate” of bacteria causing severity of malaria, where *Bacteroides* species are required components of the conglomerate. We have added a new section to the manuscript (line 330-375) where it is shown that gut *Bacteroides* are crucial in mediating the severity in the conglomerate of bacteria. These new results demonstrate the presence of *Bacteroides* does not implicitly mean mice will be susceptible to hyperparasitemia. Yet, *Bacteroides* are always present in mice susceptible to hyperparasitemia, and we have shown that treatment with individual *Bacteroides* species is sufficient to cause hyperparasitemia-resistant mice to become susceptible to hyperparasitemia. Additionally, the newly provided meta-analyses of previously published studies, identifies other potential bacteria in the conglomerate. Consequently, we believe our data demonstrate a causal role of *Bacteroides* within a conglomerate of bacteria towards susceptibility to severe malaria.
- 2) We fully agree with the reviewer that the human data does not support the conclusion that *Bacteroides* caused the differential malaria outcomes in the Ugandan children. The value of the human studies is in the demonstration that the causal role of *Bacteroides* demonstrated in the murine model has relevance to African children. Indeed, that is exactly what the human data provide to this study. That we show in the murine model *Bacteroides* species cause susceptibility to severe malaria, coupled with the correlating increase in *Bacteroides* species in Ugandan children with SMA, is perhaps the most exciting and compelling aspect of the study. Regarding the possibility that *Plasmodium* infection could change the gut microbiome, we have published that longitudinal stool sampling in Kenyan children within two weeks before and within two weeks after a febrile malaria episode that there were no changes in stool bacteria compositions (Mandal RK, et al., *Journal of Infectious Disease*, 2019). Therefore, it is possible that *Plasmodium falciparum* infections in the Ugandan children did not alter gut bacteria populations. However, multiple studies in mice and non-human primates have shown changes in gut bacteria during acute malaria and into convalescence. We have shown that, Py hyperparasitemia-resistant and -susceptible mice showed changes in gut bacteria communities following Py infection, with decreasing differences in gut bacteria communities between these groups of mice at convalescence (up to day 60 post-Py infection) compared to differences observed pre-Py infection (Denny JE, et al., *Scientific Reports*, 2019, 9:3472). To test the effect of Py-induced changes in gut bacteria, we performed FMTs from convalescent Py hyperparasitemia-resistant and -susceptible mice (day 60-post infection) into germ-free mice. Ex-germ-free mice colonized with gut microbiota from convalescent Py hyperparasitemia-resistant and -susceptible mice were resistant and susceptible, respectively, to hyperparasitemia following Py infection. These results demonstrate that, at least within this model, Py-induced changes in gut bacteria composition do not change susceptibility to future Py infections. This outcome supports the use of case-control studies (e.g., asymptomatic versus severe malaria anemia) as we report in this manuscript because it demonstrates the bacteria that are present prior

to *Plasmodium* infection and cause susceptibility to severe malaria are not lost in the susceptible mice nor do these bacteria appear in the resistant mice. Therefore, even if *P. falciparum* causes changes in human gut bacteria populations, these data suggest that comparing children with asymptomatic *P. falciparum* infections compared to children with severe malaria has the potential to identify bacteria that contributed to the differential malaria outcomes. The challenges and limitations of the human data is now acknowledged in lines 296-305.

Major points:

Line 62: The ceca was a focus. The authors should explain why they chose the ceca in particular. As noted in lines 80-83, transfer of ceca content is sufficient to transfer the resistant and susceptible phenotypes to germ-free mice. These results indicate that bacteria in the ceca are sufficient to determine susceptibility to severe malaria.

Line: 95 (Figure 1F-I), it is not clear how the data was analyzed to conclude that “all five groups had significantly different microbial compositions”. What statistics were used to conclude this? Furthermore, in line 98, it is stated that “broad taxonomic analyses do not distinguish Py hyperparasitemia-resistant and -susceptible mice”. I cannot seem to reconcile these two seemingly conflicting phrases, one stating that all groups were significantly different than one another” and the other stating that they could not be distinguished.

Thank you for bringing these confusing points to our attention. The text has been revised in lines 105-107 to indicate that Bray-Curtis beta diversity analysis revealed significant differences between the groups of mice. The statistical tests used in these analysis (PERMANOVA) is noted in the figure legend.

Line 101: Since functional profiling was done of the metagenome, how did the authors determine that they had sufficient read depth to do this analysis and was it consistent across their samples?

Since the number of genes in mouse gut jumped from ~2.6 million genes (PMID: 26414350) to > 5.8 million genes (PMCID:PMC8544893) with advancement in sequencing and bioinformatics approached, we hypothesize the numbers non-redundant genes will continue to grow. We followed a similar sequencing depth of 5GB/sample as done by Xiao et al. *Nature Biotechnology* (2015), 1103-1108 (PMID: 26414350). Thus, believe samples were sequenced to sufficient depth. Similar number of sequencing depth were obtained per sample (Figure 1C).

Line 106: The authors indicate that there is “distinct clustering of all five groups of mice except for Taconic” but in Figure S Figure 2A, it looks like Env is also overlapping with Tac-S, in addition to Tac-R overlapping with Tac-S as mentioned by the authors. So in fact, it would appear that just 2 groups are actually distinct clusters on their own (Jax and CR). This should be clarified.

Line 114-116 of the revised application clarifies this point. “Functional profiling of metagenomes at the pathway level using HUMAnN2 showed distinct clustering of CR and Jax mice with overlap between Env and Tac-S and a few shared samples between Tac-S and Tac-R (S Figure 2A).”

Line 154: It would be helpful to have more clarify on why “whole blood bulk metatranscriptomics” was done in this study. From the wording of metatranscriptomics, I assumed this was geared towards studying microbial transcripts in the blood, but I believe that this is focused solely on host transcripts, which would be better termed “whole blood transcriptomics”. That being said though, it still doesn’t quite explain why focusing on the whole blood (and presumably therefore, immune cells) would help to test the hypothesis stated in line 151 (“that Py hyperparasitemia susceptible mice upregulate the heme metabolic pathway due to enriched gut metagenomic potential for alpha-hemolysin/cyclolysin transport systems and increased capacity of gut bacteria for heme biosynthesis pathways”). The connection between the activity of the gut microbes and the immune cell transcripts is unclear and does not appear to test the hypothesis stated.

We agree the original title of this subsection along with the rationale and hypothesis for these studies were not clear. The title of the subsection has been revised (line 167). We have also revised the rationale and hypothesis (lines 168-174).

Line 172: This is an interesting hypothesis raised about bacterial interactions with red blood cells. However, is there variability in the level of bacterial translocation / gut leakiness in these different animals from different vendors? I wonder whether the leakiness factor could be more important than the gut microbial composition if the idea is that the bacteria are lysing red blood cells.

We have previously reported that both Py hyperparasitemia-resistant mice (Tac) and -susceptible mice (CR) show mild and transient increases in intestinal permeability (Denny JE, et al., *Scientific Reports*, 2019, 9:3472). We have not investigated bacterial translocation in these groups of mice. The reviewer raises an interesting possibility regarding differential bacteria translocation and the effect this may have on red blood cells, however these go beyond the scope of the current study.

Line 266: The connection to the human studies seems unclear. The idea of the mouse studies focuses on whether healthy mice having certain types of gut microbes impacts their level of parasitemia upon malaria infection. However, in the human study, the gut microbes are studied in children with different levels of disease severity from malaria. Since this is not a longitudinal study in humans that parallels what is done in mice, how do the authors rule out that the effect is not the other way around in humans (the severe malaria disease itself causing gut microbial changes?)

The relevance and importance of the human data are discussed in detail above in response to hesitation #2.

Line 299: The authors study the relatedness of the Bacteriodes species using phylogeny of the 16S gene, but this doesn’t really seem sufficient to understand the level of similarity between these species. Are there other ways to look more thoroughly into the differences between B.t. versus the other B. species?

Certainly, whole genome sequencing can be performed to find differences in the genome. Additionally, how these bacteria behave in each enterotype can be studied using

metatranscriptomics. However, these go beyond the scope of current manuscript, but it will be important to study their functional differences to gain much deeper insight.

Discussion: It would be helpful for the authors to compare their results with a 2023 publication by Farinella et al (<https://www.frontiersin.org/articles/10.3389/fcimb.2022.1058926/full>) and comment on whether results are similar or different. For example, Farinella et al describes an increase in the genetic capacity for tryptophan biosynthesis in the gut microbes during malaria infection. Figure S2B in this paper seems to indicate a similar finding in terms of significance for the tryptophan biosynthesis pathway, and may be worth commenting on.

Comparison of our results and Farinella, et al., are included in line 480-483 of the revised manuscript.

Discussion: Was a complete fecal transfer done to test the hypothesis that the complex of bacteria mediates disease severity? This seems like a good way to test that stated hypothesis. Yes, complete ceca content transfer was done from Py hyperparasitemia-resistant mice (Tac) and -susceptible mice (CR) into germ-free. Py infection of the colonized ex-germ-free mice resulted in parasitemia profiles similar to Tac and CR mice. See line 333-335.

Methods: What is the expected result of the heme assay kit when applied to lysed red blood cells? The authors indicate that they lysed red blood cells in water prior to measuring heme, but was the goal to measure heme inside of RBCs or free heme? This part could be better clarified. The goal was to measure total heme.

Methods: How were the stool samples collected in the humans? (this is not explained in the paper). Although they cite another publication, it may be helpful to briefly explain here. These details are now provided in line 744-747.

Minor

Line 23: P. malariae is misspelled
Thank you for noting this error, which has been corrected.

Line 591: PlasmoDB is misspelled
Thank you for noting this error, which has been corrected.

Line 622: May be useful to any relevant IRB approval numbers to this section.
Provided in lines 736-742 of the revised manuscript.

Reviewer #3 (Remarks to the Author):

This manuscript outlines a multiomics approach to investigate how specific species of Bacteroides may be causally linked to the risk of severe malaria. They found that isolates of Bacteroides caccae, Bacteroides uniformis, and Bacteroides ovatus but not Bacteroides

thetaiotaomicron caused susceptibility to severe malaria in mice. However, the pathogenic potential of gut *Bacteroides* towards susceptibility to severe malaria appears to be dependent on additional gut microbiota, indicating a consortium effect in severe malaria.

In general, this paper is well-written. The overall scientific approach is reasonable albeit lacking in some critical details, as outlined below. The authors appear to be quite knowledgeable in this field and have related previous publications that are relevant. Specific comments are listed below:

We appreciate the supporting comments and insightful suggestions. In particular, the importance in demonstrating beyond potential functional differences that there are actual functional differences among the gut microbiota communities in the different groups of mice. To this end, we have added new data that identify differentially abundant metabolites and metabolic pathways. Among those, we report increased abundance of metabolites shown to be produced by *Bacteroides* species.

1. The manuscript title is over-reaching and is not supported by the content of the manuscript – please revise this to be more accurate.

We believe the inclusion of the new metabolomics data (S Figure 3-4 and S Figure 11) and bacterial interaction studies (Figure 7) further support an essential role for *Bacteroides* in causing susceptibility to severe malaria. Therefore, we have not revised the title.

2. Page 1 – lines 16-18; this statement is very concerning and somewhat undermines the basic tenet of this manuscript. If the “pathogenic potential of gut *Bacteroides* is dependent on the (functions) of additional gut microbiota,” then their approach based on metagenome sequencing is too limited.

The referenced statement has been replaced (lines 18-21) in the revised abstract based on new data we provide. Among the new data, ceca and serum metabolomics data (S Figure 3-4 and S Figure 11) addresses the concern with the limited nature of metagenome sequencing. These data demonstrate functional differences among the gut microbial communities in the Py hyperparasitemia-susceptible and -resistant groups of mice. Moreover, they revealed metabolites previously shown to be produced by *Bacteroides* species (S Figure 11) were abundant in the susceptible mice.

3. Much of the text in pages 4 and 5 might be more appropriate in the Method section as opposed to Results.

We respectfully disagree with the suggestion. We believe these data are appropriate in the results to establish the overall model of the study and provide an explanation of the metagenomic analyses that were used.

4. Page 4, line 72 – this sub-header is confusing – what does “enriched genomic potential” mean? This is too vague.

Thank you for identifying the confusion in the sub-header. It has been revised (line 75) to state “microbial metagenomic potential”.

5. The authors correctly point out (page 5, line 98) that “broad taxonomic analyses do not distinguish disease phenotypes,” but they fall into a similar very problematic assumption that “more abundant genes in the genome” suggest function and then link that to a particular pathway. The authors are to be severely cautioned here – there is now copious evidence from numerous publications that predicting actual function from “metagenomic potential” is very dangerous and has led to numerous confounding results. As the authors point out, the actual metabolic activities of the gut microbiota are very context dependent and thus are best measured perhaps by metaproteomics and metabolomics. Why didn’t the authors pursue these more informative systems? At present, they metagenome “potential predictions” provide very little substantive activity information and in fact could be highly misleading.

This is an excellent point in which we fully agree. To address this concern, we have provided global metabolomics analysis from ceca of Py hyperparasitemia-susceptible mice (CR and Env) and Py hyperparasitemia-resistant mice (Tac and Jax). These are discussed in line 126-133 of the revised manuscript. Ceca metabolomics show that there are functional differences between the groups of susceptible and resistant mice (S Figure 3) with some metabolites related to *Bacteroides* (S Figure 11).

6. Page 7, line 129 – here and elsewhere are beautiful examples of where metabolomics could reveal some critical insights into actual “function” rather than “potential function.” Sad that there was a missed opportunity.

Please see response to point #5.

7. Page 8, line 154 – it is confusing what the authors mean by “whole blood bulk metatranscriptomics?” This looks to be mouse transcriptomics – where is the “metatranscriptomics?”

We apologize for the confusion of this statement. The title of the subsection has been revised to “Whole blood transcriptomics...” (line 167) to accurately reflect the data that are provided.

8. Page 9, line 182 – this section is nice but basically shows that some culturable gut bacteria increase parasite burden – how is that related to the complex multi-species function in the actual mouse gut ecosystem? As such, this is an interesting but quite disconnected.

We agree with the reviewer that these data reveal some culturable bacteria can increase parasite burden, and that the mouse (and human) gut is a complex multi-species ecosystem. None of the studies and conclusions in this report are meant to minimize this complexity. Indeed, data provided in the new Figure 7 highlights the complex interactions that likely are at hand among the gut bacteria that cause susceptibility to severe malaria. Yet, we also believe in the value of taking a reductionist approach to identify specific members of this complex ecosystem that contribute to severity of malaria. There was no guarantee this approach would work. Yet, as we report and as noted in the comment above, we were indeed able to identify individual bacteria that are part of the consortium of bacteria responsible for modulating the severity of malaria in mice. We anticipate the inclusion of the new Figure 7 places the findings of this manuscript into context of the complex gut ecosystem.

9. Page 11, line 217 – this section is focused on cultured bacteria and may totally miss other bacteria that are casual agents or co-members necessary for infection. These types of “culture studies” are valuable but do not adequately reflect the context-dependent complexity of the actual in-vivo systems. The authors need to be more clear to qualify their approach and findings and acknowledge the limitations. Otherwise, they are finding some bacterial types that can cause hyperparasitemia but this might not translate accurately into what is actually happening/controlled in the in-vivo systems.

As noted in response to point #8, we fully agree with the limitations of the culturomics approach we employed. We note the limitations of this approach in lines 206-208 of the revised manuscript. This concern is also addressed through the inclusion of the new Figure 7, which places the findings of specific bacteria within the context of the complex gut ecosystem.

10. Page 13, line 273 – the authors are to be cautioned here – linking the abundance of bacteria at the genus level is very broad – there are likely many members within that genus that are highly different in function. Thus, this classification at a quite broad level and implicated by abundance at the “genomic potential function level” has been and is likely to continue to lead to confounding information.

We agree, which is why we also investigated and report on individual *Bacteroides* species.

11. Page 14, line 302 – this statement is over-reaching and reflects some basic inherent problems with the experimental approach of this work. To make this statement based primarily on metagenomic potential function and culturable bacteria is too limited – there could be (maybe likely are) other unculturable members that are actually much more active in the disease process. This needs to be accurately acknowledged in the Discussion. Even so, the authors confess that “context matters” in that *Bacteroides* spp. are “sometimes harmful in the GI tract” – line 310.

We believe the data, including the new metabolomics data (S Figure 3 and 11) and consortium/network data (Figure 7) further support the critical role of *Bacteroides* in susceptibility to severe malaria. Yet, we also fully agree with the reviewer that inherent limitations of our approaches could also preclude us from identifying other equally important gut bacteria. We have made note of this limitation in lines 388-392.

12. Page 20, final section – it is not clear what the technical value of this manuscript is? Certainly, some gut *Bacteroides* are common among hyperparasitemia-susceptible mice but some of that information was already known. How this relates to the actual context-dependent actual metabolic activities in the in-vivo mouse is unclear? This is apparently why the authors refer in numerous places in the manuscript that their data “may imply or suggest...” Even though they tout a multi-omics approach, many of their techniques are too distant from monitoring the actual metabolic activities in the in-vivo system and thus their results seem disconnected from their goals.

We would like to thank the reviewer for raising these thought-provoking questions. Regarding the technical value of this manuscript and what is already known in the literature, we would like to highlight that:

- 1) Presently it is unknown whether gut bacteria, viruses, archaea, fungi, etc. effect malaria outcomes. These data identify a central role for gut bacteria (while not excluding potential roles for viruses, archaea, fungi, etc.).
- 2) Presently it is not known whether gut microbiota cause susceptibility or resistance to severe malaria. These data identify gut bacteria cause susceptibility rather than resistance.
- 3) The specific members of gut microbiota that effect malaria outcomes are not known. The present data explicitly delineates *Bacteroides* as a causal contributor to susceptibility to severe malaria in mice. Importantly, this manuscript also demonstrates the plausible contribution of *Bacteroides* towards severe malaria in Ugandan children that had not previously been explicitly identified. These observations in the Ugandan children have spurred on new areas of investigation with these and other human stool samples that are beyond the scope of this manuscript.

Owing to the excellent suggestion of the reviewer to move beyond functional potential in the original manuscript, we now show functional differences through inclusion of ceca metabolomics analysis. These data reveal not only global changes in ceca metabolites, but importantly increased abundance of specific metabolites associated with *Bacteroides*. This allows us to move beyond distant associations between these causal bacteria towards proximal metabolites that can be explored as mechanistic links between *Bacteroides*, altered host immune responses to Py, and severe malaria. These are also new areas of active investigation within the lab that are beyond the scope of this current manuscript.

REVIEWERS' COMMENTS

Reviewer #1 (Remarks to the Author):

The authors have addressed my comments.

Reviewer #2 (Remarks to the Author):

I appreciated the authors responses and felt that many of the items mentioned were adequately addressed, however, a few issues remain.

(1) I noted that the title remained the same, and I still did not find sufficient evidence for this statement of causality in the paper. Even just looking at the title and the abstract reveals a discordance. The title states that there is a “Causal role of gut Bacteroides in susceptibility to severe malaria” while in the abstract the authors state that “Bacteroides alone are insufficient to cause susceptibility to hyperparasitemia” and in the discussion they state “Pathogenicity of Bacteroides spp. do not satisfy Koch’s postulates to establish a causal relationship between gut Bacteroides and susceptibility to severe malaria”. Is it possible that there is a different bacterial clade that is often co-occurring with Bacteroides but is itself the cause? This possibility cannot be ruled out. I therefore would recommend the title be changed to “Association of gut Bacteroides in susceptibility to severe malaria”. I fully agree that the specific Bacteroides species shown in the paper are a “common denominator” as the authors state in the conclusion, but the causality is in question and should not remain in the title of the paper.

(2) The section now titled “Whole blood transcriptomics identified alpha toxin producing bacteria increase parasitemia” is still quite hard to understand. From the title, it is not quite clear what the method or conclusion is for this section. I think the study was performed on host transcripts but the title still suggests that it focuses on bacteria and the term “metatranscriptomics” still appears in line 174. There are also some leaps made here in terms of the idea that upregulation of heme metabolism in the host is a response to alpha-toxin-producing bacteria, which collectively result in modulating parasite burden. I agree with the authors who state, appropriately in line 200, that “further studies are needed” but this leaves me a bit unclear on the overall result of this section and why it argues that “whole blood transcriptomics identified an alpha-toxin producing bacteria that increases parasitemia” since all of these connections are not explicitly shown.

Reviewer #3 (Remarks to the Author):

The authors have now responded appropriately to most of the reviewer comments. The expanded text and addition of metabolomics data have greatly strengthened the manuscript and many of the weaker points in the previous version. While the revision is not perfect and somewhat misses the spirit of some of the minor concerns, in total this manuscript is much improved and of adequate quality now.

Point-by-Point Response to Reviewers' Comments

Reviewer #1 (Remarks to the Author):

The authors have addressed my comments.
We appreciate the support of this reviewer.

Reviewer #2 (Remarks to the Author):

I appreciated the authors responses and felt that many of the items mentioned were adequately addressed, however, a few issues remain.

(1) I noted that the title remained the same, and I still did not find sufficient evidence for this statement of causality in the paper. Even just looking at the title and the abstract reveals a discordance. The title states that there is a “Causal role of gut *Bacteroides* in susceptibility to severe malaria” while in the abstract the authors state that “*Bacteroides* alone are insufficient to cause susceptibility to hyperparasitemia” and in the discussion they state “Pathogenicity of *Bacteroides* spp. do not satisfy Koch’s postulates to establish a causal relationship between gut *Bacteroides* and susceptibility to severe malaria”. Is it possible that there is a different bacterial clade that is often co-occurring with *Bacteroides* but is itself the cause? This possibility cannot be ruled out. I therefore would recommend the title be changed to “Association of gut *Bacteroides* in susceptibility to severe malaria”. I fully agree that the specific *Bacteroides* species shown in the paper are a “common denominator” as the authors state in the conclusion, but the causality is in question and should not remain in the title of the paper.
In recognition of the consortium effect, the title has been revised to “Gut *Bacteroides* act in a microbial consortium to cause susceptibility to severe malaria”.

(2) The section now titled “Whole blood transcriptomics identified alpha toxin producing bacteria increase parasitemia” is still quite hard to understand. From the title, it is not quite clear what the method or conclusion is for this section. I think the study was performed on host transcripts but the title still suggests that it focuses on bacteria and the term “metatranscriptomics” still appears in line 174. There are also some leaps made here in terms of the idea that upregulation of heme metabolism in the host is a response to alpha-toxin-producing bacteria, which collectively result in modulating parasite burden. I agree with the authors who state, appropriately in line 200, that “further studies are needed” but this leaves me a bit unclear on the overall result of this section and why it argues that “whole blood transcriptomics identified an alpha-toxin producing bacteria that increases parasitemia” since all of these connections are not explicitly shown.

The title of this section has been further revised to “Whole blood transcriptomics and ceca metagenomics identified alpha toxin producing bacteria increase parasitemia”. We believe this will help frame the reader towards the dual use of both of these omics platforms to support this conclusion. Thank you for pointing out the inclusion of “metatranscriptomics”. The text has been revised to replace “metatranscriptomics” with “transcriptomics”. We agree with the reviewer that this particular aspect of the study requires further investigation. As a result, we avoid drawing strong conclusions from this data set. Nonetheless, it provides an association between host cell transcripts and alpha-toxin-producing bacteria that are worth sharing with the scientific community.

Reviewer #3 (Remarks to the Author):

The authors have now responded appropriately to most of the reviewer comments. The expanded text and addition of metabolomics data have greatly strengthened the manuscript and many of the weaker points in the previous version. While the revision is not perfect and somewhat misses the spirit of some of the minor concerns, in total this manuscript is much improved and of adequate quality now.

We appreciate the support of this reviewer.